# Non-uniform distribution of dendritic nonlinearities differentially engages thalamostriatal and corticostriatal inputs onto cholinergic interneurons

Osnat Oz[1], Lior Matityahu[1], Aviv Mizrahi-Kliger[1,2], Alexander Kaplan[1,2], Noa Berkowitz[1], Lior Tiroshi[1†], Hagai Bergman[1,2,3], Joshua A Goldberg[1]*

[1]Department of Medical Neurobiology, Institute of Medical Research Israel – Canada, The Faculty of Medicine, Jerusalem, Israel; [2]Edmond and Lily Safra Center for Brain Sciences, The Hebrew University of Jerusalem, Jerusalem, Israel; [3]Department of Neurosurgery, Hadassah Medical Center, Jerusalem, Israel

*For correspondence:
joshua.goldberg2@mail.huji.ac.il

Present address: †Department of Neurobiology, Northwestern University, Evanston, United States

**Abstract** The tonic activity of striatal cholinergic interneurons (CINs) is modified differentially by their afferent inputs. Although their unitary synaptic currents are identical, in most CINs cortical inputs onto distal dendrites only weakly entrain them, whereas proximal thalamic inputs trigger abrupt pauses in discharge in response to salient external stimuli. To test whether the dendritic expression of the active conductances that drive autonomous discharge contribute to the CINs' capacity to dissociate cortical from thalamic inputs, we used an optogenetics-based method to quantify dendritic excitability in mouse CINs. We found that the persistent sodium (NaP) current gave rise to dendritic boosting, and that the hyperpolarization-activated cyclic nucleotide-gated (HCN) current gave rise to a subhertz membrane resonance. This resonance may underlie our novel finding of an association between CIN pauses and internally-generated slow wave events in sleeping non-human primates. Moreover, our method indicated that dendritic NaP and HCN currents were preferentially expressed in proximal dendrites. We validated the non-uniform distribution of NaP currents: pharmacologically; with two-photon imaging of dendritic back-propagating action potentials; and by demonstrating boosting of thalamic, but not cortical, inputs by NaP currents. Thus, the localization of active dendritic conductances in CIN dendrites mirrors the spatial distribution of afferent terminals and may promote their differential responses to thalamic vs. cortical inputs.

## Editor's evaluation

This manuscript addresses the cellular and dendritic physiology of cholinergic interneurons in the striatum. The authors use a creative integration of electrophysiology and optical methods to investigate this distinctive cell type, which is critically important at the intersection of motivated behavior and disease. They uncover a mechanism through which two separate active conductances – the hyperpolarization-activated h-current (HCN) and the persistent sodium current (NaP) – act in concert to selectively boost synaptic input from the thalamus onto proximal dendrites of cholinergic interneurons.

## Introduction

The striatal cholinergic interneuron (CIN) is a key modulator of the striatal microcircuitry, impacting neuronal excitability, synaptic transmission and synaptic plasticity of spiny projection neurons (SPNs),

 

as well as other striatal interneurons (**Abudukeyoumu et al., 2019**; **Assous, 2021**; **Goldberg et al., 2012**; **Matityahu et al., 2022**). There are two processes that drive the ongoing release of acetylcholine (ACh) by CINs. First, the two glutamatergic inputs arising from cerebral cortex and intralaminar nuclei of the thalamus can drive CINs to discharge (**Bradfield et al., 2013**; **Ding et al., 2010**; **Doig et al., 2014**; **Kosillo et al., 2016**; **Lapper and Bolam, 1992**; **Mamaligas et al., 2019**; **Matsumoto et al., 2001**; **Sharott et al., 2012**; **Thomas et al., 2000**). While the unitary synaptic currents generated by these two inputs in CINs are identical (**Aceves Buendia et al., 2019**), thalamic inputs to CINs dominate in the sense that they that give rise to larger excitatory post-synaptic potentials (EPSPs) in acute striatal slices (**Johansson and Silberberg, 2020**) and can trigger abrupt pause responses, often flanked by excitatory peaks, to external saliency-related cues (**Apicella et al., 1991**; **Goldberg and Reynolds, 2011**; **Graybiel et al., 1994**; **Kimura et al., 1984**; **Matsumoto et al., 2001**; **Morris et al., 2004**; **Raz et al., 1996**). In contrast, cortical inputs to CINs are weaker in that, in acute striatal slices, they give rise to smaller EPSPs in most CINs (**Mamaligas et al., 2019**) and cannot trigger the pause response (**Ding et al., 2010**), although, these differences are less pronounced in intact animals (**Doig et al., 2014**).

Second, even in the absence of afferent input, CINs exhibit multiple autonomously generated discharge patterns – including regular and irregular pacemaking, as well as burst firing (**Bennett et al., 2000**; **Bennett and Wilson, 1999**; **Goldberg and Reynolds, 2011**; **Goldberg and Wilson, 2010**). These firing patterns are generated by an interplay between various nonlinear ionic currents including voltage- and $Ca^{2+}$-activated $K^+$ currents, as well as two voltage-dependent pacemaker currents: the hyperpolarization-activated cyclic nucleotide-gated (HCN) current; and the persistent $Na^+$ (NaP) current (**Bennett et al., 2000**; **Deng et al., 2007**; **Goldberg et al., 2009**; **Goldberg and Wilson, 2010**; **Goldberg and Wilson, 2005**; **McGuirt et al., 2021**; **Oswald et al., 2009**; **Song and Surmeier, 1996**; **Wilson, 2005**; **Wilson and Goldberg, 2006**). The main purpose of these pacemaker currents presumably is to guarantee the ongoing release of ACh onto the striatal microcircuitry by sustaining the autonomous discharge of CINs. Nevertheless, these pacemaker (and other subthreshold) currents will also impact how cortical and thalamic inputs are integrated by the CINs.

Attaining a mechanistic (e.g. dynamical systems) understanding of how the repertoire of CIN firing patterns is generated, requires a full characterization of the nonlinear properties of the pacemaker (and other) currents (e.g. by determining their voltage dependence and kinetics) – which is a daunting endeavor. In contrast, understanding how these currents impact synaptic inputs is a simpler task. Because individual synaptic inputs are small, the membrane nonlinearities can be *linearized* making the analysis of their impact simpler and more general – a treatment called the quasi-linear membrane approximation. This analysis dates back to Mauro (**Koch, 1984**; **Mauro et al., 1970**) and has shown that quasi-linear membranes can give rise to two qualitatively different transformations of inputs: *amplification* and *resonance* (**Goldberg et al., 2007**; **Hutcheon and Yarom, 2000**). Being a linear approximation, the quasi-linear approximation is amenable to Fourier analysis, which helps to better define these transformations as linear (time-invariant) filters on the input in frequency space.

Amplification arises from regenerative ionic currents – that provide positive feedback – including inward (depolarizing) currents activated by depolarization, such as the NaP current. Here, the main effect in frequency space is amplification of the amplitude response (as compared to the response of a passive linear membrane). Resonance arises from restorative currents – that provide negative feedback – including inward currents activated by hyperpolarization, such the HCN current. In frequency space, the defining properties of resonance is a peak (at a non-zero frequency) in the amplitude response, and a zero-crossing of the phase delay (at a nearby frequency). On a practical level, the quasi-linear response properties of a membrane can be measured by providing a small-amplitude, sinusoidally-modulated voltage command to an intracellularly-recorded neuron and recording the resultant sinusoidal output current. Calculating the ratio of the voltage amplitude to the current response yields an estimate of the membrane's impedance (which is, loosely speaking, an indication of the neuron's 'input resistance' to a sinusoidal input as a function of its frequency). Membrane impedances have been reported for various neuronal types in the brain (**Hutcheon et al., 1996**; **Ulrich, 2002**), including in the striatum (**Beatty et al., 2015**). **Beatty et al., 2015** found that CINs exhibit a resonance in the vicinity of 1 Hz. Moreover, they found that the shape of the impedance function depended on the holding voltage, which is understandable given that the amplitude of the various subthreshold currents is voltage-dependent. Finally, with use of tetrodotoxin (TTX), a selective antagonist of voltage-activated

Na[+] (Nav) channels, Beatty and collaborators (**Beatty et al., 2015**) demonstrated that NaP currents contributed to these filtering properties of the CIN membrane.

Use of somatic voltage perturbations, however, fails to discriminate the role played by the CINs' dendritic arbor per se in transforming synaptic inputs. The CINs' dendritic arbor can span well over half a millimeter from the soma (**Wilson, 2004**), with cortical inputs terminating on distal dendrites and thalamic inputs terminating perisomatically and on proximal dendrites (**Doig et al., 2014**; **Lapper and Bolam, 1992**; **Mamaligas et al., 2019**; **Thomas et al., 2000**). Both dendritic morphology (as taught by cable theory) and dendritic nonlinearities will lead to distal cortical inputs being integrated differently from proximal thalamic input. Thus, the impact of membrane nonlinearities on the quasi-linear approximation will depend on *where* they are expressed throughout the dendritic arbor. To address this question, we recently developed an optogenetics-based experimental method (that relies on the use of quasi-linear cable theory and Fourier analysis) to determine the duration of the delays introduced by dendrites, and how these delays impact the rapidity and fidelity of a neuron's response to its input. We used our method to study GABAergic neurons of the substantia nigra pars reticulata (SNr) that expressed channelrhodopsin-2 (ChR2). We illuminated (with a 470 nm LED) either a small perisomatic region or the entire dendritic arbor. Comparison of the two illumination regimes enabled us to demonstrate that dendrites (that in the SNr can be >700 μm long) introduce a significant integration delay. The analysis also yielded that SNr dendrites behaved like passive linear filters, without evidence for amplification or resonances, or any dependence on holding voltage (**Tiroshi and Goldberg, 2019**).

In the present study, we apply our method to study nonlinearities of CIN dendrites. We demonstrate that HCN and NaP currents shape the quasi-linear response properties of CINs, and that dendrites contribute additional phase delays. Furthermore, we show that our analysis can reveal information about dendritic location of membrane nonlinearities, and use the analysis to deduce that both HCN and NaP currents are expressed primarily proximally. Blocking NaP currents pharmacologically revealed that only perisomatic illumination triggers a boosting response. The proximal distribution of NaP currents is further supported by measuring: (a) how far autonomously generated backpropagating

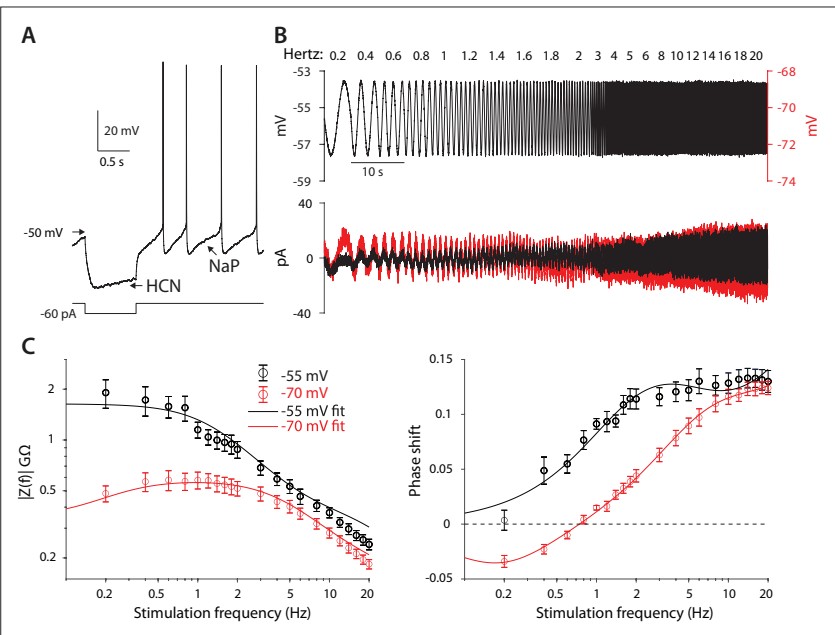

**Figure 1.** CIN membranes exhibit voltage-dependent quasi-linear properties. (**A**) CINs exhibit a voltage sag due to HCN currents, and autonomous pacemaking due to NaP currents. (**B**) Application of 2 mV sinusoidal voltage commands to the soma, of increasing frequencies, elicits a current response that is voltage dependent (black trace, –55 mV; red trace, –70 mV). (**C**) Estimation of the impedance (left) and phase shift (right) show that at –55 mV, CINs exhibit an amplified impedance and that at –70 mV, CINs exhibit a resonance (non-monotonic impedance and negative phase delays). Solid lines are parameter fits for $\left(\alpha^2 + \beta^2\right)^{-1/2}$ up to a scale factor (left), and phase shift, $\phi_s$ (right, see **Equation 5**).

action potentials (bAPs) actively invade the dendritic arbor; and (b) the differential boosting by NaP currents of proximal thalamostriatal *vs.* distal corticostriatal EPSPs.

## Results

### Ionic currents underlying amplification and resonance in cholinergic interneurons

The HCN and NaP currents depolarize CINs over largely non-overlapping voltage ranges. The HCN current is mostly active below –60 mV and is responsible for the voltage sag in response to a hyperpolarizing current pulse (*Figure 1A*, 'HCN'), whereas the NaP current takes over at –60 mV and is necessary and sufficient (*Bennett et al., 2000*) to drive CINs to action potential threshold (*Figure 1A*, 'NaP'). Therefore, because NaP is a regenerative current, while HCN is a restorative current, we would expect the current responses to an oscillating voltage command to depend strongly on whether the membrane voltage is clamped above or below –60 mV (*Beatty et al., 2015*). We therefore held mouse CINs in whole-cell voltage clamp (n=*10* neurons, N=4 mice), first at –55 mV, and subjected them to a voltage command that was composed of a continuous sequence of sinusoidal cycles with an amplitude of 2 mV and a frequency that increases discretely from 0.2 to 20 Hz. The current amplitude was very small at low frequencies and increased monotonically to the high frequency (*Figure 1B*, black), which is suggestive of an impedance curve with amplified lower frequencies. Loosely speaking, this means that the CINs' 'input resistance' to a low-frequency oscillatory input currents is boosted. As expected (*Figure 1C*, black), the impedance curve, |Z(f)|, exhibited an amplifying structure, with the phase delay being strictly positive. In order to quantify the degree of amplification, we fit the model of the phase delay, $\phi_s$, for an iso-potential cell with quasi-linear properties (see *Equation 5* in Materials and methods). In this fit, there is a (negative) amplifying parameter, $\mu_n$ – that is derived from biophysical properties of amplifying current (e.g. the slope of the activation curve, reversal potential, etc.) (*Goldberg et al., 2007*) – which was estimated to be $\mu_n = –3.9$.

When neurons were held at –70 mV (n=*10* neurons, N=7 mice), the current response was very different. It was much larger at subhertz frequencies as compared to the experiment at –55 mV, and then exhibited what looks like a slightly decreasing amplitude for frequencies near 1 Hz, followed by an amplitude increase at higher frequencies (*Figure 1B*, red). Estimation of the amplitude response and phase delay (*Figure 1C*, red), revealed significantly different curves (amplitude: p=*6·10^{–25}*, phase:

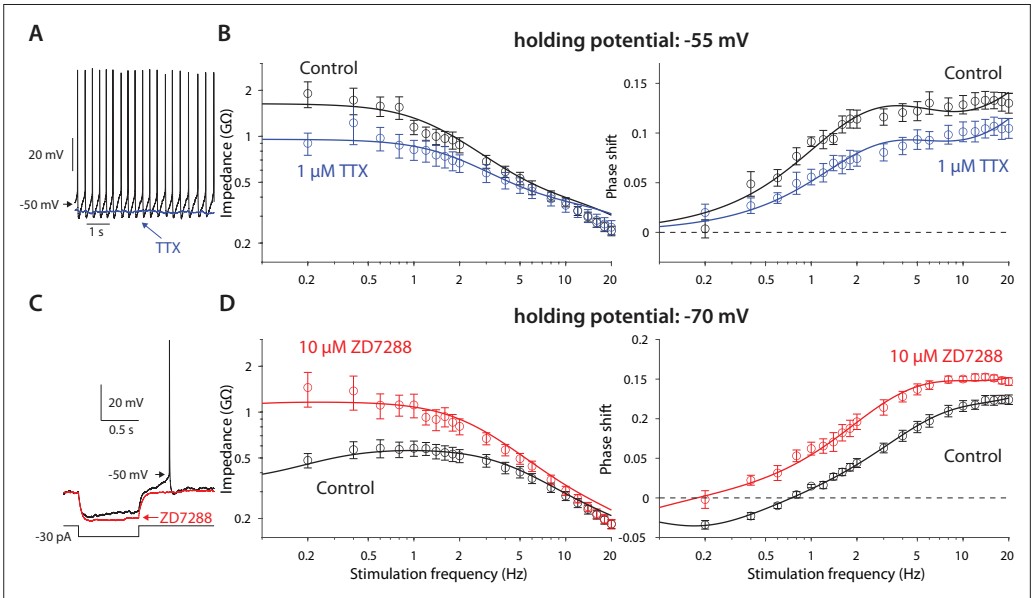

**Figure 2.** Amplification is caused by NaP currents, whereas resonance is caused by HCN currents. (**A**) TTX (blue) prevents autonomous spiking (black). (**B**) TTX prevents amplification of the impedance (left), and reduces the phase shift (right). (**C**) ZD7288 (red) abolishes the voltage sag (black). (**D**) ZD7288 abolishes the resonance peak in the impedance, and the negative phase shifts in the subhertz range. Solid lines are parameter fits as in *Figure 1*.

$p=1\cdot10^{-17}$, ANCOVA) with a clear resonance peak at approximately 1 Hz, and a zero crossing at a slightly lower frequency (*Figure 1C*, red). Fitting the somatic phase delay yielded a reasonable fit only when both an amplifying and resonant component were included in the fit with the amplifying parameter ($\mu_n = -3.4$) being only slightly reduced relative to the –55 mV fit. In contrast, the additional (positive) resonance parameter, $\mu_h$ – which is derived from the biophysical properties of the restorative current – was estimated to be $\mu_h=1.6$ (see Materials and methods) based on the fit to the negative lobe in the phase delay.

The previous experiments suggest that the amplifying effect at –55 mV occurs due to prominence of the amplifying NaP current in that voltage range, whereas the resonance visible at –70 mV is due to the HCN current that dominates that voltage range. This conclusion was supported by the fact that application of 1 µM TTX (n=9 neurons, N=4 mice), that abolished autonomous spiking and slightly hyperpolarized the CIN (*Figure 2A*), reduced the impedance ($p=9\cdot10^{-9}$) and exhibited a trend toward a shortened phase delay (*P=0.137*, ANCOVA, *Figure 2B*). These changes are captured by the amplification parameter being estimated as less negative ($\mu_n = -2.0$), which reduces $\phi_s$ (*Equation 5* in Materials and methods). Similarly, application of 10 µM ZD7288, the selective HCN antagonist (n=9 neurons, N=7 mice), which abolished the sag response (*Figure 2C*), abolished the resonance peak in the impedance curve ($p=4\cdot10^{-16}$) and significantly reduced the negative lobe in the phase delay (*p=0.018*, ANCOVA, *Figure 2D*), which was captured by the resonance parameter being reduced to $\mu_h=0.5$.

## Optogenetic interrogation of the spatial distribution of NaP and HCN currents in the CIN dendritic arbor

While the above experiments demonstrate the NaP and HCN currents are capable of transforming subthreshold voltage fluctuations, the question remains as to where within the CIN's somatodendritic compartments these currents perform their amplifying and restorative actions, respectively. One extreme scenario is that they are restricted to the axosomatic region (where they are needed for sustaining the autonomous firing patterns of the CINs). In that scenario, the dendrites could be entirely linear, passively transmitting the distal depolarizations to the soma. Only at the soma are the inputs then transformed by these currents. However, a more realistic scenario is that these currents are also expressed dendritically and exert their nonlinear influence on synaptic inputs more distally. But in this scenario, we would still want to know where along the dendrites the currents are expressed. To determine the precise location of these channels would require advanced in situ molecular and anatomical techniques and/or direct electrophysiological recordings from CIN dendrites in conjunction with advanced imaging techniques. While experiments may, in principle, be done, we wondered whether the optogenetics-based technique that we recently developed to interrogate the role of dendrites in synaptic integration (*Tiroshi and Goldberg, 2019*) could help us address this question.

We previously showed that the impact of dendrites on synaptic integration can be quantified by studying the response of neurons that express ChR2 post-synaptically to illumination of their dendritic arbor. In particular, we compared between two spatial patterns of illumination of the somatodendritc arbor of SNr neurons: either a small perisomatic region (approximately 100 µm in diameter) or the entire dendritic arbor. By using sinusoidally-modulated blue (470 nm) LED illumination at various temporal frequencies, we were able to calculate the phase delays produced by both spatial patterns, and found that illumination of the entire dendritic arbor introduced larger phase delays. In order to quantify the effect, we fit the data to a tractable theoretical model of a semi-infinite cable (*Goldberg et al., 2007*; *Tiroshi and Goldberg, 2019*). As mentioned above, in the case of the SNr neuron, we found that the dendrites were well-fit by a passive linear cable model, whose parameters (i.e., time and space constants) we could estimate. The conclusion that SNr dendrites were largely linear was further supported by the finding that these phase delays were voltage-independent (*Tiroshi and Goldberg, 2019*).

Because CINs exhibit prominent amplifying and resonating currents that are strongly voltage dependent (*Figures 1 and 2*), we posited that by using the same optogenetic technique and semi-infinite cable model we would be able to quantify the contribution of CIN dendrites to post-synaptic integration. In particular, we hypothesized that by fitting a *quasi-linear* cable model, we would be able to quantify to what degree CIN dendrites per se possess amplifying or resonating properties (See

Materials and methods). Finally, by comparing illumination of the proximal *vs.* the entire dendritic arbor, we could learn something about the localization of the nonlinearities along the dendritic arbor. To this end, we crossed mice that express Cre-recombinase under a choline acetyltransferase (*Chat* gene) promoter with the Ai32 mouse that expresses ChR2 and EYFP in a Cre-dependent manner (see Materials and methods). The cholinergic neuropil and individual CINs could be clearly visualized in the dorsal striatum of these ChAT-ChR2 mice (*Figure 3A*). Individual CINs were patched and recorded in voltage clamp, while illuminating either the proximal region with a 60 X water-immersion objective (*Figure 3A*) or the entire slice with a 5 X air objective, with a continuous sequence of sinusoidally modulated illumination waveforms at various frequencies (*Figure 3B*, blue).

Comparison of the somatic current traces in response to proximal (*Figure 3B*, black) *vs.* full-field (red) illumination demonstrated that the phase of the full-field-generated current is delayed relative to the proximally-generated current. This could be observed in the raw data both for the low and high (yellow inset) frequencies. The effect at the high frequency was very evident when plotting the phase delay curves both when the CINs were held at –55 mV (*Figure 3C*) and at –70 mV (*Figure 3D*). Estimation of these phase delays revealed delays that were considerably larger than those observed with the electrical somatic stimulation (*Figure 1*). The main contributor to the large delays are the kinetics of the ChR2, $\phi_C$ (*Equation 6b*, in Materials and methods) with an additional dendritic delay $\phi_d$ (*Equation 2* in Materials and methods). So we used independent measurements of the ChR2 kinetics in CINs (*Figure 3—figure supplement 1*), and previous literature about these kinetics (*Nagel et al., 2003*; *Tchumatchenko et al., 2013*) to fit the phase and amplitude contribution of ChR2, as explained in Appendix 1.

We found that fitting our model of ChR2 kinetics to the significantly-different phases observed in the proximal and full-field illuminations at –55 mV (n=5 neurons, N=4 mice, p=$4 \cdot 10^{-3}$, ANCOVA) yielded that the full-field illumination activated roughly twice the electrotonic range (r=1.07) that was activated by the proximal illumination (r=0.46, *Figure 3C*). However, a closer look at these fits reveals that the phase delay at 0.4 Hz (for both the proximal and full field illumination), is not captured by this model. This dip in phase delay (*Figure 3C*, green arrow) at this low frequency is reminiscent of what a restorative current is expected to do. In order to accentuate the effect of the restorative HCN currents, we repeated the experiment at the –70 mV holding potential (*Figure 3D*). In this case (n=14 neurons, N=10 mice), the phase delays at the lower frequencies (especially at 0.4 Hz) were negative, which was reminiscent of the results from the electrical voltage stimulation experiments (*Figure 1*) at the –70 mV holding potential. Estimation of the amplitude responses for proximal and distal stimulation at both holding potentials revealed that they were less sensitive at revealing the resonance structure (*Figure 3—figure supplement 2*), that was more readily read off from the phase responses (*Figure 3D*) in the sense that phase estimates provided tighter error bars than the amplitude estimates.

When fitting a full ChR2 kinetics plus quasi-linear dendrite model (incorporating both amplifying and restorative parameters) to the –70 mV measurements (*Figure 3D*), we found that the curves were significantly different (p=$8 \cdot 10^{-4}$, ANCOVA) and the estimates of the effective electrotonic range of illumination remained similar to those estimated with the passive model (*Figure 3C*): r=1.06 for the full-field and r=0.58 for the proximal illumination. Here too the restorative parameter was estimated to be larger for the proximal fit ($\mu_h$=3.4) relative to the full field fit ($\mu_h$=1.7), suggesting that the HCN currents are denser more proximally.

*Table 1* summarizes the model parameters used to fit the curves in *Figures 1–3*. In Appendix 1, we discuss how the various parameters affect the model and provide a more detailed description of how the parameter space was searched to fit the model.

The conclusion that the HCN current is concentrated proximally is buttressed by the fact that the negative phase delays in the subhertz frequency range tended to be less negative for the full-field illumination (P=0.037, Wilcoxon rank-sum test on phase delays at 0.4 Hz). In the framework of our quasi-linear model, the only mathematically possible way to recreate this finding while increasing the effective electrotonic range r being illuminated is by reducing the restorative parameter $\mu_h$. Otherwise, if $\mu_h$ were constant then illuminating a larger region of the cable – by increasing r – would necessarily recruit more restorative current, thereby making the negative phases more negative (See Appendix 1).

Aside from the deviation at 0.4 Hz (*Figure 3C*, green arrow, which may be indicative of a resonant component), the other phase measurements at –55 mV are consistent with a simple model of ChR2

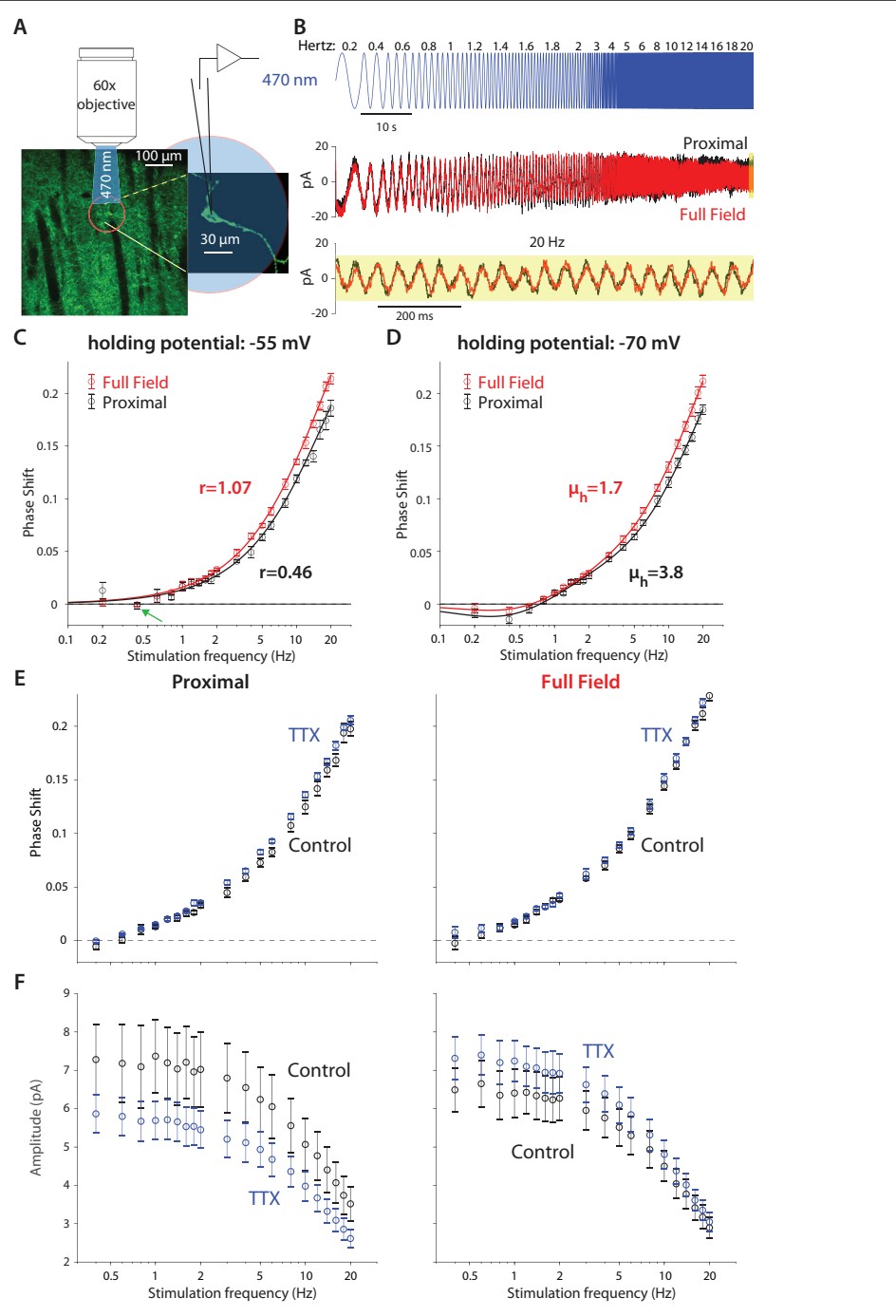

**Figure 3.** Optogenetic interrogation of the quasi-linear properties of CIN dendrites indicates that dendritic nonlinearities are more prominent proximally. (**A**) A CIN in a sagittal slice from ChAT-ChR2 mouse is patch-clamped in the whole-cell mode while either a small proximal region around the soma or the full-field are illuminated with a sinusoidally modulated 470 nm LED. (**B**) The current response to the proximal (black) and full-field (red) illumination differ, with the phase of the full-field illumination right-shifted at the higher frequencies (20 Hz is highlighted in yellow). (**C**) Phase shifts at −55 mV holding potential, calculated for proximal (black) and full-field illumination (red). A tendency towards negative phase shifts is present at 0.4 Hz (green arrow). Fitting the passive model at −55 mV demonstrated that the effective range of illumination (**r**) is larger for the full field fit (*Equation 2*). (**D**) Phase shifts at −70 mV holding potential, exhibit a negative phase shift, and the resonance parameter ($\mu_h$) is smaller for the full field fit, as is the magnitude of the amplification parameter ($\mu_n$, see main text). The elevation in these parameters' magnitudes when illuminating proximally relative to full-field suggests that the surface densities of NaP and HCN

*Figure 3 continued on next page*

*Figure 3 continued*

currents are higher proximally. (**E**) Phase response for proximal (left) and full-field illumination (right) in TTX at –55 mV. (**F**) Amplitude response for proximal (left) and full-field illumination (right) at –55 mV reveals an opposite effect of TTX.

The online version of this article includes the following figure supplement(s) for figure 3:

**Figure supplement 1.** High-frequency phase delays in response to optogenetic activation are attributable to ChR2 kinetics and dendritic delays.

**Figure supplement 2.** Amplitude responses to proximal (black) and full-field (red) illumination at the two holding potentials.

kinetics plus a passive dendritic delay, thereby raising the possibility that optogenetic dendritic activation fails to engage NaP currents. This failure could suggest that NaP currents are relatively absent from the dendrites. However, our somatic measurements showed clear evidence that NaP currents affect the quasi-linear properties of the soma (*Figure 2B*). Additionally, the fit to the phases at –70 mV, required the inclusion of an amplification parameter, which was estimated to have a larger magnitude for the proximal illumination ($\mu_n = -1.5$) than for the full-field illumination ($\mu_n = -0.9$). It therefore seems more likely that the Nav channels that underlie the NaP currents are localized proximally and taper off distally. To test this, we repeated the measurements at –55 mV before and after application of TTX. While TTX application may have slightly increased the phase delays for both proximal and full-field illumination (*Figure 3E*) – which could reflect a reduction in the overall dendritic membrane conductance of the CINs, and therefore a lengthening of its space constant – the amplitude responses (*Figure 3F*) unequivocally show that TTX exerts an *opposite* affect when proximal *vs.* full-field illumination are used. While full-field illumination in TTX increased the somatic current's amplitude response (control: n=9 neurons, N=7 mice; TTX: n=14 neurons, N=8 mice, p=7·10⁻⁹, ANCOVA) presumably by reducing the current escape via the dendritic membrane, proximal illumination in TTX reduced the somatic current's amplitude response (control: n=9 neurons, N=7 mice; TTX: n=12 neurons, N=6 mice, p=1.1·10⁻⁴, ANCOVA) indicating that proximal NaP currents indeed boost the somatic current, as concluded from the somatic experiments (*Figure 2*). Thus, this pharmacological result provides *model-independent* evidence that the NaP currents are expressed primarily proximally and less so distally. In the following section, we provide independent evidence in support of this conclusion.

## Distance of dendritic bAP invasion indicates location of amplifying Nav channels

The persistent and fast-inactivating Nav currents flow through the same Nav channels (*Alzheimer et al., 1993*). Therefore, a method that is indicative of where these NaV channels are located will indicate where the NaP current can be found. One such method involves determining to what distance from the soma dendritic bAPs invade the dendritic arbor. To this end, we used 2PLSM Ca²⁺ imaging to measure the size of the Ca²⁺ transients elicited by bAPs in autonomously firing CINs at various distances from the soma (*Figure 4A*). We conducted line scans (n=11 neurons, N=7 mice) to measure the Ca²⁺ signals at various distances from the soma (*Figure 4B*). Next, we estimated the size of the spike

**Table 1.** Parameters fit to quasi-linear model in Figures 1–3.

| Figure, Curve | $\mu_n$ | $\mu_h$ | $\tau_n$ (ms) | $\tau_h$ (ms) | $\gamma_R$ | $\tau$ (ms) | $r$ | Amp (GΩ) |
|---|---|---|---|---|---|---|---|---|
| 1 C, –55 mV | –3.9189 | - | 28 | - | 5.4296 | 41.8 | - | 2.4728 |
| 1 C, –70 mV | –3.3854 | 1.5863 | 11.6 | 1077.8 | 5.8722 | 26.1 | - | 1.4229 |
| 2B, TTX | –2.0178 | - | 31.7 | - | 3.4610 | 19.4 | - | 1.3809 |
| 2B, ZD7288 | –1.8988 | 0.4844 | 13.4 | 9107.5 | 2.6111 | 17.2 | - | 0.8323 |
| 3 C, proximal | - | - | - | - | - | 35.8 | 0.4360 | - |
| 3 C, distal | - | - | - | - | - | 26.8 | 1.0651 | - |
| 3D, proximal | –1.4763 | 3.4400 | 60.0 | 551.2 | 3.3405 | 33.6 | 0.5788 | - |
| 3D, distal | –0.8885 | 1.6646 | 49.2 | 403.3 | 3.8850 | 45.3 | 1.0646 | - |

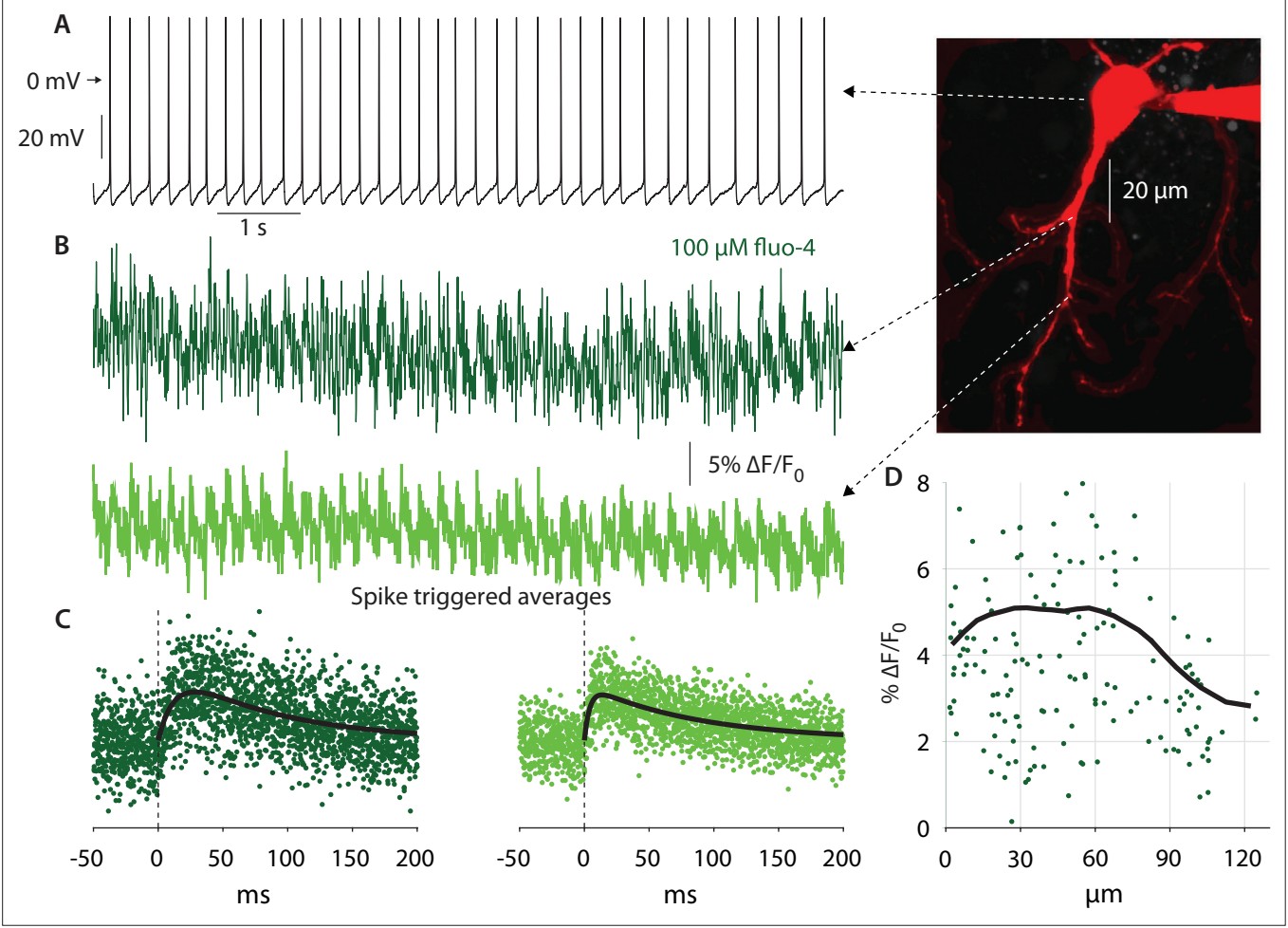

**Figure 4.** Autonomous action potentials actively back-propagate in CIN dendrites up to 70 μm from the soma. (**A**) Autonomous discharge of a CIN that was patched and filled with fluo-4 and Alexa Fluor 568 for 2PLSM imaging (image). (**B**) Line scans at various distances from the soma exhibit Ca²⁺ oscillations caused by bAPs. (**C**) Calculating the spike-triggered average of these oscillations and fitting an alpha-function gives an estimate of the amplitude of these oscillations (in % $\Delta F/F_o$). (**D**) The scatter plot of these amplitudes as a function of distance from the soma (11 CINs from 7 mice are pooled) exhibits a large degree of variability. However, a 35 μm moving average (black line) exhibits that the Ca²⁺ transients begin to decay approximately 70 μm from the soma, indicating that bAPs are supported by Nav channels up to that point (the initial increase up to 20 μm is due to the increase in the surface-to-volume ratio in the large proximal dendrites).

triggered average (STA) of the $\Delta F/F_0$ Ca²⁺ signal, by averaging around spontaneous APs measured at the soma (*Figure 4C*). The amplitude of the STA was estimated by fitting it with an alpha-function (see Materials and methods). A scatter plot of STA amplitude *vs.* distance from soma demonstrates a large degree of variability (*Figure 4D*). Nevertheless, applying a 35 μm moving average to the scatter plot reveals a trapezoidal dependence of Ca²⁺ transient sizes (*Figure 4D*, black). We previously demonstrated that somatic Ca²⁺ transients are smaller than dendritic transients due to the difference in the surface-area-to-volume ratio (*Goldberg et al., 2009*; *Rehani et al., 2019*). In the same vein, the initial dip in the size of the transients at short distances from the soma result from the large size of the proximal dendrites (*Figure 4D*), relative to the distal dendrites. Neglecting that effect, we found that the size of the bAP-driven Ca²⁺ transients remains constant up to approximately 70 μm from the soma, and then drops off. Additionally, in 7 CINs (N=5 mice) in which we had measurements of $\Delta F/F_0$ at both proximal and distal (>70 μm) locations, we found that the median distal signal was significantly lower than the proximal one by 29% (p=*0.047*, Wilcoxon rank-sum test). This spatial dependence of the Ca²⁺ transient amplitude, suggests that the bAP maintains a constant amplitude due to the presence of Nav channels that sustain their regenerative nature out to some 70 μm from the soma. Farther out, the Ca²⁺ transients decrease presumably due to a drop off in Nav channel expression, which leads to a lower

amplitude bAPs, and hence less Ca²⁺ via voltage-activated Ca²⁺ channels. Thus, this measurement strengthens the conclusion that NaP currents are present in CIN dendrites – primarily in proximal dendrites (up to approximately 70 µm from the soma).

## NaP currents boost proximal thalamic inputs but not distal cortical inputs to CINs

In a previous study of CINs in the Q175 mouse model of Huntington's disease (HD), we found that their total NaP current is larger, and that bAPs invade farther out into their dendrites (up to 120–130 µm from the soma) in the Q175 mice relative to wildtype mice. In that study, we also found that optogenetically-activated corticostriatal excitatory postsynaptic currents (EPSCs) in CINs were boosted by NaP in these HD mice, because they were reduced by ranolazine, a selective antagonist of the NaP current (*Tanimura et al., 2016*). The facts: (a) that wildtype mice have smaller NaP currents (*Tanimura et al., 2016*) that are concentrated more proximally (*Figures 2B and 3F*); and (b) that bAPs invade CIN dendrites effectively only up to 70 µm from the soma (*Figure 4*) – raise the question as to whether distal cortical inputs are boosted in CINs from wildtype mice. To test this, we used Thy1-ChR2 mice (*Aceves Buendia et al., 2019*; *Arenkiel et al., 2007*; *Matityahu et al., 2022*) that expresses ChR2 in cortical fibers (*Figure 5A*), but not in the parafascicular nucleus (PfN) of the thalamus (*Figure 5—figure supplement 1A*) or in the pedunculopontine nucleus (*Gradinaru et al., 2009*), another nucleus that provides monosynaptic glutamatergic projection to CINs (*Assous et al., 2019*). Interestingly, we found in acute striatal slices from these wildtype mice that ranolazine failed to reduce the amplitude of monosynaptic corticostriatal EPSPs (*Figure 5—figure supplement 1B*) to CINs (n=11 neurons, N=6 mice, p=0.74, Wilcoxon signed-rank test, *Figure 5B*), indicating the cortical inputs to CINs are not normally boosted.

Taken together with the previous findings in the HD mice (*Tanimura et al., 2016*), this finding

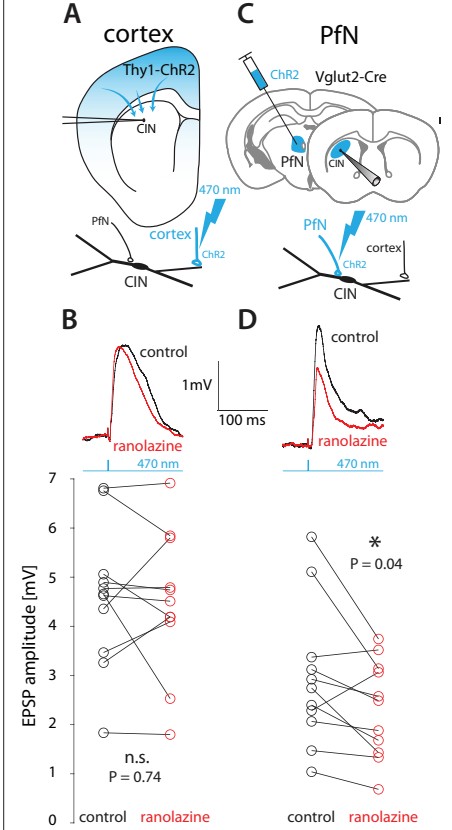

**Figure 5.** Thalamic – but not cortical EPSPs – onto CINs are boosted by NaP currents in wildtype mice. (**A**) CINs were patched in current clamp in Thy1-ChR2 mice, so that 470 nm LED illumination of striatal slices activated nominally cortical terminals. (**B**) Optogenetically evoked monosynaptic cortical EPSPs were unaffected by ranolazine. (**C**) The parafascicular nucleus (PfN) of Vglut2-Cre mice was inoculated with AAVs harboring Cre-dependent ChR2, so that 470 nm LED illumination of striatal slices activated monosynaptic PfN terminals while CINs were patched in current clamp mode. (**D**) Optogenetically evoked thalamic EPSPs in CINs (held between –50 mV and –60 mV) were reduced by 30 µM ranolazine.

The online version of this article includes the following figure supplement(s) for figure 5:

**Figure supplement 1.** Monosynaptic excitatory cortical and thalamic paired-pulse ratios (PPRs) are not affected by ranolazine.

suggests that whether or not distal cortical inputs are boosted depends on how far into the dendritic arbor the NaP current is capable of producing amplification. In wildtype mice, where the data (*Figures 2–4*) suggest the NaP currents are restricted proximally, distal cortical inputs are not boosted. However, in HD mice where NaP currents reach farther out into the dendrites (*Tanimura et al., 2016*) cortical inputs are boosted. A corollary of this conclusion would be that PfN inputs that terminate proximally on CINs (*Doig et al., 2014*; *Lapper and Bolam, 1992*; *Mamaligas et al., 2019*; *Thomas et al., 2000*) should exhibit ranolazine-sensitive boosting even in wildtype mice, because they presumably contact proximal regions where the dendritic membrane exhibits amplification (*Figure 3F*). To

test this, we transcranially inoculated the PfN of Vglut2-ires-Cre mice with adeno-associated viruses (AAVs) harboring Cre-dependent ChR2 and EYFP genes (*Figure 5C*; *Aceves Buendia et al., 2019*; *Rehani et al., 2019*). Two weeks later, we tested the sensitivity of optogenetically evoked monosynaptic EPSPs (*Figure 5—figure supplement 1B*) to ranolazine. Indeed, the mean amplitude of the EPSP was reduced by a median of 16% (n=*11* neurons, N=*4* mice, p=*0.04*, Wilcoxon signed-rank test, *Figure 5D*). Importantly, ranolazine had no effect on paired-pulse ratios (*Aceves Buendia et al., 2019*) at either cortical or PfN synapses (*Figure 5—figure supplement 1C*), ruling out a presynaptic mechanism-of-action. Thus, we conclude that the post-synaptic boosting of glutamatergic inputs to CINs by NaP currents occurs in dendritic regions that correspond to the spatial localization of these currents as derived from the quasi-linear properties of CINs and from 2PLSM imaging of bAPs.

## Tonically active neurons in non-human primates exhibit a pause-like response to slow wave oscillations during sleep

Our data, alongside previous studies (*Beatty et al., 2015*), demonstrate that somatodendritic HCN currents give rise to resonances in the subhertz range in CIN surface membranes. In contrast, SPNs exhibit no resonance (*Beatty et al., 2015*). This suggests CINs in intact animals may exhibit increased sensitivity to oscillatory inputs in that frequency range, whereas SPNs should not. Delta waves in the electroencephalogram (EEG) and in cortical and sub-cortical local field potentials (LFP) are prominent during non-REM sleep across species (*Brown et al., 2012*; *Liu and Dan, 2019*), including in non-human primates (NHPs) (*Mizrahi-Kliger et al., 2018*). Because the LFP is widely thought to represent subthreshold cellular activity, affected by afferent and recurrent synaptic inputs (*Buzsáki et al., 2012*), we hypothesized that the tonically active neurons (TANs) in the NHP striatum – that are comprised primarily of CINs (*Aosaki et al., 1995*; *Kawaguchi, 1993*; *Wilson et al., 1990*) – will exhibit stronger entrainment to slow-wave LFP events during sleep, while SPNs under the same conditions will not. Indeed, triggering the spike trains of TANs and SPNs (from N=*2* NHPs) on slow wave events that occur during sleep stages N2 and N3 (*Mizrahi-Kliger et al., 2018*; *Riedner et al., 2007*), which represent relatively deep non-REM sleep, demonstrates that while the firing rate of SPNs was unaffected (n=*83*), the firing rates of TANs (n=*122*) was modulated by these slow-wave events (*Figure 6A–E*). In contrast, the TANs firing was not modulated during higher frequency sleep spindle events (*Figure 6F–H*), in agreement with the subhertz membrane resonance peak we observed in the acute slice experiments. Importantly, the biphasic response exhibited by TANs to slow-wave events bears a striking resemblance to the 'classical' TAN response to external cues (*Aosaki et al., 1995*; *Apicella et al., 1991*; *Joshua et al., 2008*; *Kimura et al., 1984*). Because these responses in awake primates require an intact thalamic projection (*Bradfield et al., 2013*; *Goldberg and Reynolds, 2011*; *Matsumoto et al., 2001*; *Schulz and Reynolds, 2013*; *Smith et al., 2004*; *Yamanaka et al., 2018*) (and there is no reason to assume that this is altered in sleep), our findings (*Figure 6*) provide support to the hypothesis that the CIN membrane resonance contributes to a thalamically-driven biphasic CIN response during slow-wave sleep, possibly because this projection terminates proximally on CINs. Importantly, because the primates are asleep in a sound-proof room, this is, to the best of our knowledge, the first report of striatal TANs responding to an internally generated brain event, and not an external saliency-related cue.

## Discussion
### Quasi-linear dendritic properties of CINs and the non-uniform distribution of their active conductances

In the current study, we applied a cable-theoretic and optogenetics-based formalism – that we developed previously to study dendritic properties of SNr neurons (*Tiroshi and Goldberg, 2019*) – to study the dendritic properties of CINs. Unlike SNr neurons that were found to act like passive linear cables exhibiting no voltage dependent properties, we find that CIN dendrites exhibit both amplification and resonances in a voltage dependent manner: amplification is more prominent at depolarized subthreshold potentials (approximately –55 mV), whereas resonances are more prominent at more hyperpolarized potentials (approximately –70 mV). Moreover, we found that our method is able to provide information about where in the dendritic arbor these quasi-linear properties are localized. We extracted boosting ($\mu_n$) and resonance ($\mu_h$) parameters for proximal *vs.* full-field illumination, and found that the magnitude of both parameters is smaller when the entire dendritic arbor is illuminated.

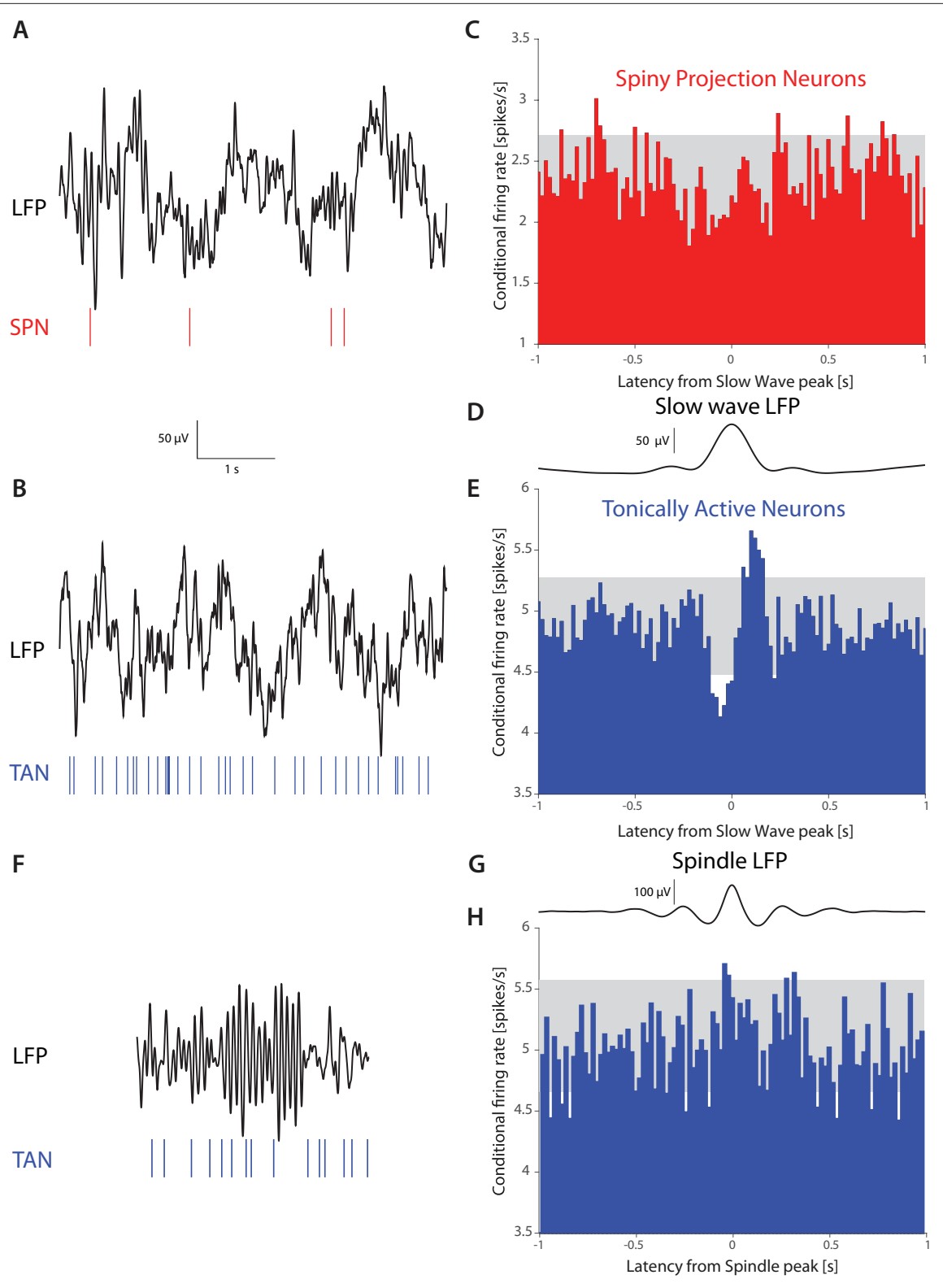

**Figure 6.** TANs, but not SPNs, exhibit a pause-like response to slow-wave events occurring during natural non-REM sleep in non-human primates (NHPs). (**A**) Simultaneous recording of LFP and an SPN in an NHP during N2 and N3 stages of sleep. (**B**) Simultaneous recording of LFP and a TAN in an NHP during N2 and N3 stages of sleep. (**C**) SPN firing rate conditioned on the occurrence of an LFP slow wave event (6,065 triggers). (**D**) Average striatal LFP signal triggered on the occurrence of slow wave events (see Materials and methods). (**E**) TAN firing rate conditioned on the occurrence of slow wave event (28,603 triggers). (**F**).Simultaneous recording of LFP and a TAN in an NHP during a sleep spindle. (**G**) Average striatal LFP signal triggered on the

*Figure 6 continued on next page*

*Figure 6 continued*

occurrence of a sleep spindle (see Materials and methods). (**H**) TAN firing rate conditioned on the occurrence of a sleep spindle (5829 triggers). Gray box indicates the 99% confidence intervals.

In other words, activating more of the distal dendritic membrane dilutes the amplifying and resonating effects, meaning that the additional membrane illuminated in the full-field condition contributed less to amplification and resonances. The simplest interpretation of this finding is that the active conductances that give rise to amplification and resonance are more highly expressed proximally, and apparently taper off farther out in the dendritic arbor. We identified the relevant currents by studying the quasi-linear properties of CIN somata with voltage perturbations. We found that amplification is TTX-sensitive and therefore arises from Nav channels that give rise to the NaP current. Similarly, we found that resonances require HCN channels.

We used one direct and two indirect methods to validate the preferentially proximal localization of NaP currents. First, TTX application revealed that boosting occurs only at the soma (*Figure 2B*) and in proximal (but not distal, *Figure 3F*) dendritic membranes. Importantly, the TTX results demonstrate that our conclusion that boosting is restricted to the soma and proximal dendrites is *independent* of the quasi-linear model fit. Second, we implemented a widely-used approach of estimating the distance of dendritic invasion of bAPs by imaging the $Ca^{2+}$ influx that accompanies them (*Carter and Sabatini, 2004*; *Day et al., 2008*; *Goldberg et al., 2009*; *Kerr and Plenz, 2004*; *Rehani et al., 2019*; *Tanimura et al., 2016*). We found – not unlike in SPN neurons – that bAPs maintain their amplitude out to 70 μm from the soma and then begin to decay, suggesting that Nav currents that support bAP propagation taper off from that point onwards. Because strictly speaking a change in $Ca^{2+}$ transients could result from a change in the concentration of voltage-activated $Ca^{2+}$ channels, future work can verify our results by directly testing where locally puffed TTX attenuates the bAP (perhaps even with the use of voltage sensitive probes). Second, we demonstrated that only thalamic – but not cortical – EPSPs exhibit sensitivity to the selective NaP blocker. Because the thalamic terminals are located more proximally (*Lapper and Bolam, 1992*; *Mamaligas et al., 2019*; *Thomas et al., 2000*), this provides further evidence that NaP currents are more prominent locally. The preferential localization of HCN currents proximally in CIN dendrites requires additional validation, particularly because other neurons exhibit an opposite pattern (*Berger et al., 2001*; *Harnett et al., 2015*; *Kole et al., 2006*). However, we were less interested in validating the proximal localization of HCN currents, because unlike NaP currents, activation of HCNs is conditioned on CINs being actively (after-)hyperpolarized, during which bAPs are unlikely to occur. We did, however, test a context in which HCN currents, and the resonance they underlie, are likely to affect CIN collective dynamics. We found that TANs respond biphasically to internally generated slow wave activity during sleep, but not to sleep spindles. This preferential entrainment is consistent with the fact that CIN membranes exhibit a resonance in the subhertz range as found by us and others (*Beatty et al., 2015*). The elevated impedance (or frequency-dependent access resistance) in this range means that inputs fluctuating in this range will more efficiently depolarize the CINs, and therefore more likely to trigger additional action potentials than inputs fluctuating at a frequency that is far from the resonance frequency. Accordingly, SPNs that do not exhibit resonances (*Beatty et al., 2015*) are not entrained by the slow wave oscillation. Our findings in sleeping NHPs are at odds with various rodent studies that found that cortical slow-wave activity is weakly associated with the discharge of TANs in anesthetized rodents, and more strongly associated with SPN activity (*Goldberg et al., 2003*; *Mahon et al., 2001*; *Reig and Silberberg, 2014*; *Schulz et al., 2011*; *Sharott et al., 2012*; *Stern et al., 1998*). These differences may be attributable to species differences and/or differences between natural sleep and anesthesia, which may differ in the degree to which thalamic *vs.* cortical inputs engage striatal neurons. Moreover, the dendritic nonlinearities in SPNs (*Plotkin et al., 2011*) may be preferentially engaged during anesthesia, thereby causing SPNs to respond more strongly under anesthesia than during natural sleep.

## Advantages and limitations of the quasi-linear formalism and the optogenetics-based approach

As mentioned in the Introduction, the quasi-linear approach provides a tool to characterize membrane excitability in general functional terms, without requiring the complete biophysical characterization of membrane currents. Unfortunately, the method is not invertible: even after estimating all the

quasi-linear parameters one cannot deduce the full biophysical characterization of the underlying channels even if one has a precise mathematical model of these channels. Moreover, even though we extracted the values of $\mu_n$ and $\mu_h$, we do not know how to translate them into a quantitative measure of channel density. Still, the ability to compare parameters for various spatial illumination patterns enables us to reach qualitative conclusions regarding relative channel density. Additionally, our attribution of $\mu_n$ to NaP and $\mu_h$ to HCN is only provisional. CINs possess other channels that have a major influence on their firing patterns such as A-type, inward rectifying, and various $Ca^{2+}$-activated $K^+$ channels, to mention a few (*Bennett et al., 2000*; *Deng et al., 2007*; *Goldberg et al., 2009*; *Goldberg and Wilson, 2010*; *Goldberg and Wilson, 2005*; *McGuirt et al., 2021*; *Oswald et al., 2009*; *Song and Surmeier, 1996*; *Wilson, 2005*; *Wilson and Goldberg, 2006*) that were not included in our analysis. Future work should elaborate how these and other channels contribute to the quasi-linear properties of CINs and their dendrites.

Using optogenetics provides another level of practicality. The seminal studies that characterized the filtering properties of dendrites or pyramidal neurons (*Berger et al., 2001*; *Goldberg et al., 2007*; *Hutcheon et al., 1996*; *Ulrich, 2002*) required dual soma-dendrite patching which is not practical for all neuronal types whose dendrites taper off rapidly. Expressing opsins in the membrane being activated, as we did, means that a single somatic patch electrode can suffice to conduct the quasi-linear characterization. We currently use this approach to illuminate large regions of the dendritic arbor simultaneously, which means we can only derive large-scale dendritic properties. In the future, localized laser stimulation of dendritic regions (visualized with the help of a fluorescent marker in the patch pipette), or even two-photon laser activation of opsins (provided one can find stimulation parameters that do not harm the dendrites) could be used as an alternative approach. On the backdrop of optogenetics being utilized almost exclusively to study circuit mapping (*Häusser, 2021*; *Kim et al., 2017*; *Petreanu et al., 2009*) by expressing opsins presynaptically, we believe our study joins other studies (*Higgs and Wilson, 2017*; *Tiroshi and Goldberg, 2019*) to underscore the value of using opsins expressed postsynaptically to study neuronal – even dendritic – excitability.

## Somatic amplification and resonance in CINs

We preceded our optogenetic characterization of dendritic nonlinearities with the electrical characterization of quasi-linear somatic properties using a sinusoidal voltage command applied through the patch pipette. As previously reported by Beatty and collaborators (*Beatty et al., 2015*), CIN somata exhibit a membrane resonance in the subhertz range, that depends on the holding potential. However, while Beatty and collaborators found that an amplified resonance is present in the more depolarized (approximately – 55 mV) range, we found that the resonance is only pronounced in the more hyperpolarized (approximately –70 mV) range, where it depends on the HCN current. It is possible that the resonances observed by Beatty and collaborators (*Beatty et al., 2015*) at –55 mV arose from ineffective clamping of dendrites with the slightly higher resistance electrodes used in that study. In support of this proposition, we also found evidence for the resonance arising from unclamped (dendritic) membranes. When the CINs were clamped at –55 mV, while somatic voltage perturbations did not produce a resonance, optogenetic stimulation (both proximal and full field) at 0.4 Hz produced a more negative phase than at the neighboring 0.2 Hz and 0.6 Hz stimulation (*Figure 3C*, green arrow). This downward deflection in phase is reminiscent of the full-blown negative subhertz region that occurred with optogenetic stimulation when the CINs were clamped at –70 mV (*Figure 3D*). Thus, it is likely that resonance originated in both studies from dendritic HCN currents. Because in the study of Beatty and collaborators (*Beatty et al., 2015*) TTX lowered the impedance, without removing the subhertz resonance peak, we conclude that the effect of TTX in both studies is to remove the NaP-dependent boosting, without directly affecting resonance. It also seems that the location of the peak amplitudes estimated by Beatty and collaborators (*Beatty et al., 2015*) is perhaps 1 Hz larger than the location of the peaks in our study. However, this difference could result from the fact that they used a chirp stimulation where the frequency increases continuously, whereas we used several periods of perfect sinusoidal waveforms with a discrete set of frequencies. Moreover, it can be shown mathematically that for the quasi-linear model the zero crossing in the phase response occurs at a slightly smaller frequency than the peak in the amplitude response, so this may also contribute to the impression that the resonance frequency observed in our experiments is slightly different from that observed in Beatty and collaborators (*Beatty et al., 2015*).

## Dendritic contribution to capacity of CINs to differentiate between their excitatory inputs

In NHPs, TANs that are comprised primarily of CINs (*Aosaki et al., 1995*; *Goldberg and Wilson, 2010*; *Kawaguchi, 1993*; *Wilson et al., 1990*), encode – through a brief pause in their tonic firing – external stimuli that are salient and often unexpected, even conveying a stop or behavioral shift signal (*Aoki et al., 2018*; *Thorn and Graybiel, 2010*) or stimuli that are associated with reward (*Apicella et al., 1991*; *Goldberg and Reynolds, 2011*; *Kimura et al., 1984*). The pause response requires an intact projection from the PfN of the thalamus (*Bradfield et al., 2013*; *Goldberg and Reynolds, 2011*; *Matsumoto et al., 2001*; *Schulz et al., 2011*; *Schulz and Reynolds, 2013*; *Smith et al., 2004*; *Yamanaka et al., 2018*), indicating that TANs are attuned to thalamic input. While TANs do not respond to ongoing movement (*Aosaki et al., 1994*; *Apicella et al., 1991*; *Kimura et al., 1984*; *Raz et al., 1996*), and hence seem less attuned to sensorimotor cortical input (*Sharott et al., 2012*), CIN activity in awake behaving mice is strongly modulated by self-initiated movements (*Gritton et al., 2019*; *Howe et al., 2019*; *Yarom and Cohen, 2011*). Nevertheless, slice physiology studies in rodents (*Aceves Buendia et al., 2019*; *Assous, 2021*; *Assous et al., 2017*; *Ding et al., 2010*; *Johansson and Silberberg, 2020*; *Kosillo et al., 2016*; *Threlfell et al., 2012*) clearly show that thalamic inputs to CINs are stronger in the sense that they give rise to larger EPSPs (*Johansson and Silberberg, 2020*) and can trigger a pause-like responses (*Ding et al., 2010*), whereas cortical inputs to most CINs (*Mamaligas et al., 2019*) are weaker in that they give rise, in acute slices, to smaller EPSPs and cannot trigger the pause-like response – although this distinction is less pronounced in anesthetized rodents (*Doig et al., 2014*). Thus, it is clear that CINs differentiate between these two excitatory inputs and respond differently to them. A major contributor to the CINs' capacity to dissociate thalamic and cortical input is the differential distribution of their respective terminals on the dendritic arbor: thalamic input terminate perisomatically and on proximal dendrites, whereas cortical input terminates on distal dendrites (*Doig et al., 2014*; *Lapper and Bolam, 1992*; *Mamaligas et al., 2019*; *Thomas et al., 2000*). Our finding that the higher expression of NaP currents in proximal dendrites preferentially boosts thalamic over cortical inputs, suggests that active conductances are expressed in CIN dendrites in a manner that corresponds and reinforces the effect of the spatial separation between the terminals of the two inputs.

## Adaptive changes in dendritic excitability in movement disorders

The capacity of the CIN dendritic excitability to mirror the distribution of afferent terminals is also an adaptive process. As mentioned above, we previously reported that CIN dendritic excitability is elevated in the Q175 mouse model of HD (*Tanimura et al., 2016*). In contrast to control (Thy1-ChR2) mice in which cortical EPSPs in CINs are insensitive to ranolazine (*Figure 5*), cortical EPSCs in CINs from Q175 mice (crossed with Thy1-ChR2 mice) are strongly attenuated by ranolazine. The acquired dependence of distal cortical inputs on NaP currents in Q175 mice results from an upregulation of the NaP current, which was also evidenced by bAPs invading farther out into the CINs' dendritic arbor (*Tanimura et al., 2016*). This elevated excitability in the Q175 mouse is probably a homeostatic response aimed at elevating the postsynaptic sensitivity to the remaining synaptic contents after the loss of afferent cortical and thalamic inputs observed in HD mouse models (*Deng et al., 2013*). Indeed, CINs more readily aquired pause responses to cortical inputs in Q175 HD mice than in wild-type mice (*Tanimura et al., 2016*). Because thalamostriatal inputs are also altered in models of Parkinson's disease (PD) (*Aceves Buendia et al., 2019*; *Parker et al., 2016*; *Tanimura et al., 2019*), future work should determine whether the excitability of CIN dendrites is altered in these models, as well. Importantly, dendrites are positioned at the interface between synaptic inputs and the intrinsic properties of CIN. While changes in synaptic transmission and intrinsic excitability of CINs have received attention in models of PD, HD and other movement disorders (*Abudukeyoumu et al., 2019*; *Aceves Buendia et al., 2019*; *Choi et al., 2020*; *Ding et al., 2006*; *Ding et al., 2011*; *Eskow Jaunarajs et al., 2015*; *Mallet et al., 2019*; *Paz et al., 2021*; *Pisani et al., 2007*; *Plotkin and Goldberg, 2019*; *Poppi et al., 2021*; *Tanimura et al., 2019*; *Tubert and Murer, 2021*), studying alterations in dendritic excitability in these models is a complementary approach that remains largely unchartered territory.

The membrane channels that are expressed in CINs (e.g. Nav, HCN, Kv4, etc.) are targets for neuromodulation under normal, healthy circumstances (*Deng et al., 2007*; *Helseth et al., 2021*; *Maurice et al., 2004*; *Song and Surmeier, 1996*). Therefore, it is likely that the dendritic excitability

of CINs can be modulated under physiological conditions, as well, as a way to adjust the tuning of CINs to their afferent inputs. Conversely, if boosting and resonances are present primarily proximally, the selective tuning of CINs to subhertz slow-wave oscillations is probably stronger for thalamic inputs (that terminate proximally) than for cortical inputs. Future experiments can address this issue by testing how silencing the PfN affects the observed entrainment of TANs to slow-waves (*Figure 6*).

In summary, we have used a new optogenetics-based approach, complemented by other electrophysiological and imaging approaches, to demonstrate that the spatial localization of active dendritic conductances that endow CIN dendrites with quasi-linear filtering properties corresponds to the spatial distribution of their two main afferent excitatory inputs. This matching up of presynaptic terminals with post-synaptic excitability, probably contributes to the capacity of CINs to respond differentially to cortical and thalamic inputs. Sensorimotor information of cortical origin seems to be integrated continuously in a moment-by-moment fashion. In contrast, thalamic inputs lead to abrupt pauses in TAN firing, even in response to internally generated brain-states (e.g. during slow-wave activity). The possibility that dendritic arbors adapt to the spatiotemporal patterns of afferent inputs is likely an important principle of neural computation that deserves further attention, particularly in the framework of autonomously active neurons such as CINs, SNr neurons and other basal ganglia pacemakers.

# Materials and methods

## Key resources table

| Reagent type (species) or resource | Designation | Source or reference | Identifiers | Additional information |
|---|---|---|---|---|
| Genetic reagent (*M. musculus*) | B6.129S-*Chat*^{tm1(cre)Lowl}/MwarJ | The Jackson Laboratory | Strain #:031661 RRID:IMSR_JAX:031661 | B6J.ChAT-IRES-Cre (Δneo) |
| Genetic reagent (*M. musculus*) | B6.Cg-Tg(Thy1-COP4/EYFP)18Gfng/J | The Jackson Laboratory | Strain #:007612 RRID:IMSR_JAX:007612 | Thy1-ChR2-YFP |
| Genetic reagent (*M. musculus*) | B6;129S-*Gt(ROSA)26Sor*^{tm32(CAG-COP4*H134R/EYFP)Hze}/J | The Jackson Laboratory | Strain #:012569 RRID:IMSR_JAX:012569 | Ai32(RCL-ChR2(H134R)/EYFP) |
| Genetic reagent (*M. musculus*) | STOCK *Slc17a6*^{tm2(cre)Lowl}/J | The Jackson Laboratory | Strain #:016963 RRID:IMSR_JAX:016963 | Vglut2-ires-cre |
| Chemical compound, drug | Mecamylamine hydrochloride | Sigma-Aldrich | Lot # 019M4108V CAS: 826-39-1 | |
| Chemical compound, drug | Atropine sulfate salt monohydrate | Sigma-Aldrich | Lot # BCBH8339V CAS No.: 5908-99-6 | |
| Chemical compound, drug | SR 95531 hydrobromide (Gabazine) | Hello Bio | CAS: 104104-50-9 | |
| Chemical compound, drug | DNQX | TOCRIS | CAS: 2379-57-9 | |
| Chemical compound, drug | D-AP5 | Hello Bio | CAS:79055-68-8 | |
| Chemical compound, drug | CGP 55845 hydrochloride | Hello Bio | CAS: 149184-22-5 | |
| Chemical compound, drug | Dihydro-β-erythroidine hydrobromide | TOCRIS | CAS: 29734-68-7 | |
| Chemical compound, drug | Ranolazine dihydrochloride | Sigma-Aldrich | CAS Number: 95635-56-6 | Product Number R6152 |
| Chemical compound, drug | ZD7288 | MedChemExpress | CAS No.: 133059-99-1 | Synonyms: ICI D7288 |
| Chemical compound, drug | Tetrodotoxin citrate | Hello Bio | CAS: 18660-81-6 | |
| Chemical compound, drug | 4-Aminopyridine | Sigma-Aldrich | CAS Number: 504-24-5 | |
| Chemical compound, drug | Phosphate buffered saline tablets | Sigma-Aldrich | MDL number: MFCD00131855 | Product Number P4417 |
| Chemical compound, drug | Paraformaldehyde | Sigma-Aldrich | CAS Number: 30525-89-4 | |

*Continued on next page*

*Continued*

| Reagent type (species) or resource | Designation | Source or reference | Identifiers | Additional information |
|---|---|---|---|---|
| Chemical compound, drug | XYLAZINE AS HYDROCHLORIDE | EUROVET ANIMAL HEALTH B.V | CAS: 082-91-92341-00 | |
| Chemical compound, drug | CLORKETAM | VETOQUINOL | CAS: 1867-66-9 | |
| Chemical compound, drug | Meloxicam | Chanelle Pharmaceuticals Manufacturing ltd | CAS Number: 71125-38-7 | |
| Chemical compound, drug | isoflurane | Primal Critical Care | CAS Number: 26675-46-7 | |
| Software, algorithm | MATLAB | MathWorks | RRID:SCR_001622 | Data analysis |
| Software, algorithm | WinWCP | University of Strathclyde Glasgow | RRID:SCR_014713 | Data acquisition |
| Software, algorithm | FEMTOSmart Software: MESc | FEMTONICS | RRID:SCR_018309 | 2 P Data acquisition |
| Software, algorithm | NIS-Elements Basic Research | Nikon instruments | RRID:SCR_002776 | Confocal images acquisition |
| Software, algorithm | Signal 6 | Cambridge Electronic Design | RRID:SCR_017081 | |
| Software, algorithm | AlphaLab SnR | Alpha-Omega Engineering | https://www.alphaomega-eng.com/ | |
| Software, algorithm | Electrode Positioning System | Alpha-Omega Engineering | https://www.alphaomega-eng.com/ | |
| Other | Model 940 Small Animal Stereotaxic Instrument with Digital Display Console | Kopf Instruments | https://kopfinstruments.com/product/model-940-small-animal-stereotaxic-instrument-with-digital-display-console/ | See "Stereotaxic viral inoculation in Vglut2-ires-Cre mice" in Materials and Methods |
| Other | Sound-attenuating room | IAC acoustics | https://www.iacacoustics.com/ | See "Non-Human Primates (NHPs)" in Materials and Methods |
| Other | Glass-coated Tungsten electrodes | Alpha-Omega Engineering | https://www.alphaomega-eng.com/ | See "Non-Human Primates (NHPs)" in Materials and Methods |

## Mice

Homozygous Ai32 (RCL-ChR2(H134R)/EYFP) mice (The Jackson laboratory [Jax] stock: 012569) that express floxed ChR2 and an EYFP fusion protein under the CAG promoter were crossed with homozygous ChAT-IRES-Cre (Δneo) mice the express Cre recombinase under the *Chat* promoter (Jax stock: 031661). The ChAT-ChR2 offspring (4–8 weeks old/both sexes) were used for the majority of experiments. To investigate corticostriatal transmission, we used homozygous transgenic Thy1-ChR2 mice (B6.Cg-Tg (Thy1-COP4/EYFP) 18Gfng/1, Jax stock: 007612), that express ChR2 under the *Thy1* promoter in cortical neurons (*Arenkiel et al., 2007*). To investigate thalamostriatal transmission, we used Vglut2-ires-Cre mice (Jax stock: 016963).

## Non-human primates (NHPs)

Data were obtained from two healthy, young adult, female vervet monkeys. The monkeys were habituated to sleeping in a primate chair, positioned in a dark, double-walled sound-attenuating room. The primate chair restrained the monkeys' hand and body movements but otherwise allowed them to be in a position similar to their natural (sitting) sleeping posture. Detailed sleep habituation, surgery and sleep staging were reported previously (*Mizrahi-Kliger et al., 2018*). For extracellular recordings, the monkeys' heads were immobilized with a head holder, and eight glass-coated tungsten microelectrodes were advanced the dorsal striatum. Electrical signals were amplified with a gain of 20, filtered using a 0.075 Hz (2 pole) to 10 kHz (3 pole) Butterworth filter and sampled at 44 kHz by a 16-bit analog/digital converter. Spiking activity was sorted online using a template matching algorithm. The striatum was identified based on its stereotaxic coordinates according to MRI imaging and primate atlas data (*Martin and Bowden, 2000*). TANs and SPNs were identified using real-time assessment of their electrophysiological features. Spiking and LFPs were recorded only for identified recording sites with stable recording quality (i.e., where single-neuron spiking yielded an average isolation score ≥ 0.85).

We followed established procedures for slow wave detection in the LFP (*Riedner et al., 2007*). Briefly, the LFP signal was filtered at the 0.5–4 Hz range and putative slow wave events whose duration was 0.25–2 s were kept for further analysis. Next, the slow wave peaks were sorted according to their amplitude. Artifacts were removed by discarding all events whose amplitude exceeded 5

standard deviations above the mean. Finally, conditional firing rate analysis was only performed on 30% of the slow wave events with the highest amplitude. Conventional procedures were also used for sleep spindle detection in the LFP (*Sela et al., 2016*). Briefly, the detection algorithm was only used for striatal sites whose LFP showed significant 10–17 Hz (spindle range) activity. LFP was filtered at the 10–17 Hz range, and the Hilbert transform was then used to extract the instantaneous amplitude. Events exceeding 3 standard deviations above the mean were deemed potential spindle events, and a threshold of one half of a standard deviation above the mean was used to detect the start and end points of an individual sleep spindle. A potential sleep spindle was defined as such only if it lasted 0.5–3 s, and provided it did not exhibit a relatively high (more than 4.5 standard deviations above the mean) amplitude in a control 20–30 Hz range. The spindle data were obtained using a 4-pole 4–25 Hz Butterworth filter.

## Histology

An 8-week-old male Thy1-ChR2 mouse was deeply anesthetized with a terminal dose of ketamine-xylazine followed by perfusion through the heart of cold PBS and 4% PFA. The removed brain was kept overnight at 4 °C in 4% PFA. The next day, the brain was washed 3 × 15 min before 50 μm coronal slices of the PfN were cut with a vibratome (Leica VT1000S). VECTASHIELD (Vector Laboratories) was applied onto mounted slices to protect from bleaching. Coronal slices of the PfN were imaged using confocal microscope (Nikon A1R) using 10 x lens and a 20 x oil immersion lens to visualize constitutive EYFP expression.

## Stereotaxic viral inoculation in Vglut2-ires-Cre mice

Mice were deeply anesthetized with isoflurane in a non-rebreathing system (2.5% induction, 1–1.5% maintenance) and placed in a stereotaxic frame (Kopf Instruments, Tujunga, CA). Temperature was maintained at 35 °C with a heating pad, artificial tears were applied to prevent corneal drying, and animals were hydrated with a bolus of injectable saline (10 ml/kg) mixed with analgesic (2.5 mg/kg Meloxicam). Stereotaxic injections into caudal intralaminar nuclei of thalamus were performed under aseptic conditions. Adeno-associated viruses (AAV) serotype 9 carrying double-floxed fusion genes for hChR2 (E123A) and EYFP under an *EF1a* promoter (University of Pennsylvania Vector Core, Addgene #35507) were used to transfect PfN neurons. Injection coordinates were from Bregma: lateral, 0.65 mm; posterior, 2.3 mm; and 3.35 mm depth from surface of brain (*Rehani et al., 2019*). A small hole was bored into the skull with a micro drill bit and a glass pipette was slowly inserted at the PfN coordinates. To minimize backflow, solution was slowly injected; a total volume of 250 nl ($>2.5 \times 10^{12}$ GC/ml) of the AAV constructs was injected over a period of approximately 1.5 min and the pipette was left in place for 5 min before slowly retracting it. Slice physiology experiments were conducted 2–3 weeks after surgery.

## Slice preparation

Mice were deeply anesthetized with ketamine (200 mg/kg)–xylazine (23.32 mg/kg) and perfused through the heart with ice-cold-modified artificial cerebrospinal fluid (ACSF) bubbled with 95% $O_2$–5% $CO_2$, and containing (in mM) 2.5 KCl, 26 $NaHCO_3$, 1.25 $Na_2HPO_4$, 0.5 $CaCl_2$, 10 $MgSO_4$, 0.4 ascorbic acid, 10 glucose and 210 sucrose. The brain was removed, and 275 μm thick sagittal slices containing the striatum were cut in ice-cold-modified ACSF. Slices were then submerged in ACSF, bubbled with 95% $O_2$–5% $CO_2$, containing (in mM) 2.5 KCl, 126 NaCl, 26 $NaHCO_3$, 1.25 $Na_2HPO_4$, 2 $CaCl_2$, 2 $MgSO_4$ and 10 glucose, and stored at room temperature for at least 1 hr prior to recording.

## Electrophysiological recording

The slices were transferred to the recording chamber mounted on a Zeiss Axioskop 60 X, 0.9 NA fixed-stage microscope and perfused with oxygenated ACSF at 31 °C. During the optogenetic stimulation experiments, in order to guarantee that the effects we measured were generated post-synaptically, the ACSF solution contained (in μM) 10 DNQX to block AMPA receptors, 50 D-APV to block NMDA receptors, 10 gabazine (SR95531) to block GABA_A receptors, 2 CGP55845 to block GABA_B receptors, 10 atropine to block muscarinic ACh receptors, and 10 mecamylamine to block nicotinic ACh receptors. In the experiments in which optogenetics were used to stimulate cortical or thalamic input we used the same blockers, except for DNQX which was left out. An Olympus 40 X, 0.8

NA water-immersion objective with a 26.5 mm field number (FN) was used to examine the slice using standard infrared differential interference contrast video microscopy. Patch pipette resistance was typically 4–5 MΩ when filled with recording solutions. The junction potential estimated at 7–8 mV was not corrected. In EPSC measurements, the intracellular solution contained (in mM): 127.5 $CsCH_3SO_3$, 7.5 CsCl, 10 HEPES, 10 TEA-Cl, 4 phosphocreatine disodium, 0.2 EGTA, 0.21 $Na_2GTP$, and 2 $Mg_{1.5}ATP$ (pH = 7.3 with CsOH, 280–290 mOsm/kg). In the $Ca^{2+}$ imaging experiments (see below) the internal solution contained (in mM) 135 K-gluconate, 5 KCl, 2.5 NaCl, 5 Na-phosphocreatine, 10 HEPES, 0.1 fluo-4 (Molecular Probes), 0.1 Alexa Fluor 568 (for morphological visualization, Molecular Probes), 0.21 $Na_2GTP$, and 2 $Mg_{1.5}ATP$, pH 7.3 with KOH (280–290 mOsm/kg). In all other experiments, the intracellular solution contained (in mM) 135.5 $KCH_3SO_3$, 5 KCl, 2.5 NaCl, 5 Na-phosphocreatine, 10 HEPES, 0.2 EGTA, 0.21 $Na_2GTP$, and 2 $Mg_{1.5}ATP$, pH 7.3 with KOH (280–290 mOsm/kg). Electrophysiological recordings were obtained with a MultiClamp 700B amplifier (Molecular Devices, Sunnyvale, CA). Signals were filtered at 10 kHz online, digitized at 10 or 20 kHz and logged onto a personal computer with the Signal 6 software (Cambridge Electronic Design, Cambridge, UK).

## Voltage-perturbation experiments

CINs were held at either –55 mV or –70 mV and were given an 83 second-long voltage command structured as a concatenated sequence of sinusoids from a discrete set of frequencies ranging from 0.2 to 20 Hz with an amplitude of 2 mV (3 or 5 s per frequency, such that each frequency was represented by an integer multiple of its fundamental period). Phase shifts between the voltage sinusoidal and the somatic current response were determined by the location of the peak in the cross-correlation function (CCF) of the two traces (whose units are mV·pA), for each stimulation frequency and for each illumination condition. The impedance at each frequency, $|Z(f)|$, was calculated from the maximal amplitude of the CCF as $|Z(f)| = (2 \text{ mV})^2 / \max(\text{CCF})$ (so that its units are GΩ).

## Optogenetic stimulation

Optogenetic stimulation was performed with blue-light (470 nm) LED illumination via the objective (Mightex, Toronto, ON, Canada). We used two spatial illumination regimes: (a) *proximal illumination* wherein an opaque disk with a central pinhole was placed in the back focal plane of the 60 X water-immersion objective such that a ~130 μm diameter region around the soma was illuminated (*Tiroshi and Goldberg, 2019*), thereby targeting the soma and proximal dendrites; and (b) *full-field illumination* of the entire slice with a 5 X air objective which excites the soma and the entire dendritic field. In all experiments, LED light intensity was chosen such that stimulation generated comparable current responses for both regimes. We used the same sequence of sinusoids described above, only this time the voltage driving the LED was modulated (the minimal voltage was the LED's voltage threshold, 40 mV). The phase delays were again calculated according to the latency of the peak of the CCF between the LED voltage command and somatic current. Note, that the phases were corrected by 0.5 (i.e., by π in radians) due to the fact that the ChR2 inward current is in antiphase with the LED's voltage command. The amplitude response was calculated from the peak value of the CCF, normalized by the amplitude of the 470 nm LED command (i.e. 1 V for proximal illumination and 0.1 V for the full-field illumination, so that its units are picoamperes).

To activate the excitatory synaptic inputs in the Thy1-ChR2 and in the Vglut2-mice a full-field 470 nm LED 1 ms-long pulses were used with GABA, ACh and NMDA receptor blockers in the ACSF. For EPSPs we average 25 trials (3 s intervals, and trials with spikes were omitted). Paired-pulse ratio (PPR) measurements consisted of 64 trials of two pulses (100ms apart, 3 s interval). The mean EPSC amplitude was calculated as the difference between the mean peak current and the mean baseline current that preceded the pulse. PPR was the ratio of the second mean EPSC to the first mean EPSC. To demonstrate that the EPSCs were monosynaptic they were recorded before and after application of 1 μM TTX and 100 μM 4-aminopyridine (4-AP) (*Petreanu et al., 2009*).

To estimate the kinetics of the ChR2 currents, brief 1ms-long 470 nM LED pulses (1 V for proximal and 0.1 V for the full-field illumination) were repeated 250 times and the resulting average current response was measured, and fit with an alpha function

$$A \left( e^{-t/\tau_r} - e^{-t/\tau_d} \right) \tag{1}$$

to estimate the $\tau_r$ and $\tau_d$, the rise and decay times, respectively.

## Two-photon laser scanning microscopy (2PLSM)

The two-photon excitation source was a Chameleon Vision II tunable Ti:Sapphire pulsed laser (Coherent, Santa Clara, CA, USA) tuned to 820 nm. The images were collected with the Femto2D system (Femtonics, Budapest, Hungary) which includes two 3 mm galvo-scanners, one gated GaAsP and one multi-alkaline non-descanned photomultiplier tube for imaging fluo-4 and Alexa Fluor, respectively. Z-stacks of optical sections (spaced 2 µm apart) were collected using 0.2 µm pixels and 15 µs dwell times. Optical and electrophysiological data were obtained using the software package MES (Femtonics), which also integrates the control of all hardware units in the microscope. The software automates and synchronizes the imaging signals and electrophysiological protocols. Data in MATLAB format was extracted from the MES package to personal computers using proprietary code (Femtonics). We recorded spontaneously occurring bAPs with line scans at various distances measured radially from the tip of the soma. Spike triggered averages of the $Ca^{2+}$ measurements ($\Delta F/F_0$) were estimated and an alpha-function (*Equation 1*) was fit to them. The value of the peak of the fitted alpha-function was used as a measure of the size of the spontaneous bAP at that location.

## Drugs and reagents

TTX was used to block voltage-activated $Na^+$ currents. Ranolazine was used to block NaP currents. ZD7288 was used to block the HCN current. Gabazine (SR-95531) and CGP 55845 were used to block $GABA_A$ and $GABA_B$ receptors, respectively. 4-Aminopyridine (4-AP) was used to enable optogenetically-driven monosynaptic release in the presence of TTX. All reagents, drugs and toxins were purchased from either Merck/Sigma-Aldrich (Darmstadt, Germany), Tocris Biosciences (Bristol, UK), MedChemExpress (Monmouth Junction, NJ, USA) or HelloBio (Bristol, UK).

## Data analysis and statistics

Data were analyzed and curve fitting was performed using the *lsqcurvefit* function in MATLAB (Math-Works, Natick, MA, USA) with the parameters listed in Appendix 1. The nonparametric two-tailed Wilcoxon signed-rank test was used for matched samples and the Wilcoxon rank-sum test was used for independent samples. The parametric ANCOVA test was used to test significant changes in the amplitude and phase curves as a function of the natural logarithm of the frequencies (an transformation that spreads out this independent parameter more uniformly). Null hypotheses were rejected if the p-value was below 0.05.

For the TAN and SPN locking to slow wave or spindle peak analysis, confidence intervals (at p-value of 0.01) were calculated based on the distribution of conditional firing rates 1 s before and after the slow wave peak.

## Parameter fitting to the phase delay

In our previous study (*Tiroshi and Goldberg, 2019*), we modeled the dendritic arbor as a semi-infinite cable with a homogeneous quasi-linear membrane (i.e. the current density of each nonlinearity is constant along the dendrite). When a segment of length *r* (measured in units of the dendrite's space constant) from the soma is activated with a sinusoidal current injection, the dendritic phase delay is given by.

$$\phi_d = \frac{1}{2\pi}\left(\tan^{-1}\frac{q}{p} - \tan^{-1}\frac{\sin qr}{e^{pr}-\cos qr}\right) \tag{2}$$

where

$$p = \sqrt{\frac{\sqrt{\alpha^2+\beta^2}+\alpha}{2}} \qquad q = \sqrt{\frac{\sqrt{\alpha^2+\beta^2}-\alpha}{2}} \tag{3}$$

$\alpha$ and $\beta$ are functions of frequency and are determined by the linearization of the dendritic nonlinearities as explained in *Goldberg et al., 2007* and *Remme and Rinzel, 2011*, with a negative amplifying parameter, $\mu_n$, and a positive resonance parameter, $\mu_h$ (See Appendix 1).

$$\alpha(f) = \gamma_R + \frac{\mu_n}{1+(2\pi f \tau_n)^2} + \frac{\mu_h}{1+(2\pi f \tau_h)^2} \tag{4a}$$

$$\beta(f) = 2\pi f \left[ \tau - \frac{\mu_n \tau_n}{1+(2\pi f \tau_n)} + \frac{\mu_h \tau_h}{1+(2\pi f \tau_h)} \right] \tag{4b}$$

Additional parameters include total dendritic conductance (relative to leak), $\gamma_R$, the membrane time constant $\tau$, and the time constants representing the kinetics of the nonlinear dendritic conductances, as explained in *Goldberg et al., 2007* and in *Remme and Rinzel, 2011*. In some cases, we only used the amplifying parameter in the fit (e.g., *Figure 1C*), and in *Figure 3C*, we used *Equation 2* in the case of a passive dendrite for which $\alpha(f) = 1$ and $\beta(f) = 2\pi f \tau$.

Using the same formalism it is easy to show that for the (isopotential) soma with a quasi-linear membrane the phase delay is given by.

$$\phi_s = \frac{1}{2\pi} \tan^{-1} \frac{\beta}{\alpha} \tag{5}$$

corresponding to an amplitude that is equal to $\left( \alpha^2 + \beta^2 \right)^{-1/2}$ up to a scaling factor. In the main text, we point out that when $\mu_n$ becomes less negative $\phi_s$ is reduced. This is because when $\mu_n$ becomes less negative, $\alpha$ is increased (*Equation 4a*) and $\beta$ is decreased (*Equation 4b*).

The amplitude attenuation and phase delays generated by the ChR2 kinetics are calculated from the Fourier transform of the alpha function (*Equation 1*) and are given by.

$$A_C(f) \propto \left\{ \left( 1 - \tau_r \tau_d (2\pi f)^2 \right)^2 + \left( 2\pi f [\tau_r + \tau_d] \right)^2 \right\}^{-1/2} \tag{6a}$$

$$\phi_C(f) = \frac{1}{2\pi} \tan^{-1} \left( \frac{2\pi f [\tau_r + \tau_d]}{1 - \tau_r \tau_d (2\pi f)^2} \right) \tag{6b}$$

## Acknowledgements

This work was funded by grants from the European Research Council (ERC) Consolidator Grant (no. 646886), the Israel Science Foundation (nos. 154/14 & 155/14), and the U.S.-Israel Binational Science Foundation (no. 2017020) to JAG and grants from the Israel Science Foundation (no. 2051/20), German Research Foundation (CRC TRR295), Israel-China Binational Science Foundation and the Silverstein Foundation to HB. We would like to thank Ronit Cherki, Tamar Licht and Anatoly Shapochnikov for excellent technical assistance.

## Additional information

### Funding

| Funder | Grant reference number | Author |
|---|---|---|
| European Research Council | 646886 | Joshua A Goldberg |
| Israel Science Foundation | 2051/20 | Hagai Bergman |
| Deutsche Forschungsgemeinschaft | CRC TRR295 | Hagai Bergman |
| Israel Science Foundation | 154/14 | Joshua A Goldberg |
| Israel Science Foundation | 155/14 | Joshua A Goldberg |
| U.S.-Israel Binational Science Foundation | 2017020 | Joshua A Goldberg |

The funders had no role in study design, data collection and interpretation, or the decision to submit the work for publication.

### Author contributions

Osnat Oz, Lior Matityahu, Alexander Kaplan, Data curation, Formal analysis; Aviv Mizrahi-Kliger, Conceptualization, Data curation, Formal analysis, Writing – original draft; Noa Berkowitz, Data curation; Lior Tiroshi, Conceptualization, Data curation, Formal analysis, Methodology; Hagai Bergman,

Conceptualization, Resources, Supervision, Funding acquisition, Validation, Methodology, Project administration, Writing – review and editing; Joshua A Goldberg, Conceptualization, Resources, Data curation, Formal analysis, Supervision, Funding acquisition, Validation, Investigation, Visualization, Methodology, Writing – original draft, Project administration, Writing – review and editing

### Author ORCIDs
Osnat Oz ⓘ http://orcid.org/0000-0002-3427-921X
Lior Matityahu ⓘ http://orcid.org/0000-0002-6115-8608
Aviv Mizrahi-Kliger ⓘ http://orcid.org/0000-0001-6492-8770
Alexander Kaplan ⓘ http://orcid.org/0000-0002-4634-4322
Noa Berkowitz ⓘ http://orcid.org/0000-0002-1313-4708
Lior Tiroshi ⓘ http://orcid.org/0000-0001-5634-4484
Hagai Bergman ⓘ http://orcid.org/0000-0002-2402-6673
Joshua A Goldberg ⓘ http://orcid.org/0000-0002-5740-4087

### Ethics
All experimental protocols were conducted in accordance with the National Institutes of Health Guide for the Care and Use of Laboratory Animals, and with the Hebrew University guidelines for the use and care of laboratory animals in research. The experiments adhered to, received prior written approval from and were supervised by the Institutional Animal Care and Use Committee of the Faculty of Medicine, under protocols: MD-16-13518-4 (HB) and MD-18-15657-3 (JAG).

### Decision letter and Author response
Decision letter https://doi.org/10.7554/eLife.76039.sa1
Author response https://doi.org/10.7554/eLife.76039.sa2

## Additional files

### Supplementary files
• Transparent reporting form

### Data availability
Tables with all data points used in the figures is available at Open Science Framework: https://osf.io/yxej3/.

The following dataset was generated:

| Author(s) | Year | Dataset title | Dataset URL | Database and Identifier |
|---|---|---|---|---|
| Goldberg J, Bergman H | 2021 | Non-uniform distribution of dendritic nonlinearities differentially engages thalamostriatal and corticostriatal inputs onto cholinergic interneurons | https://osf.io/yxej3/ | Open Science Framework, yxej3 |

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

# Appendix 1

## Fitting the quasi-linear model and opsin-mediated attenuation and phase shifts

### Modeling somatic voltage perturbations

The properties of a quasi-linear membrane are captured by the following equations (*Equation 4a*, *Equation 4b* in the Materials and Methods)

$$\alpha\left(f\right) = \gamma_R + \frac{\mu_n}{1+\left(2\pi f \tau_n\right)^2} + \frac{\mu_h}{1+\left(2\pi f \tau_h\right)^2}$$

$$\beta\left(f\right) = 2\pi f \left[\tau - \frac{\mu_n \tau_n}{1+\left(2\pi f \tau_n\right)^2} - \frac{\mu_h \tau_h}{1+\left(2\pi f \tau_h\right)^2}\right]$$

The membrane time constant is given by $\tau$; the total nonlinear conductance relative to leak is given by $\gamma_R$; the negative amplifying parameter is given by $\mu_n$, and the positive resonance parameter is given by $\mu_h$. $\tau_n$ and $\tau_h$ are the corresponding time constants of these nonlinear conductances. The notation and formalism used for describing the quasi-linear membrane is based on previous publications by us and others, where the values of the above parameters were extracted from the underlying biophysical models of the nonlinear conductances (*Goldberg et al., 2007*; *Remme and Rinzel, 2011*; *Tiroshi and Goldberg, 2019*). The reader is referred to those articles for further reading.

We begin with the passive membrane, where $\mu_n = \mu_h = 0$ and $\gamma_R = 1$. In this case the amplitude, $\left(\alpha^2 + \beta^2\right)^{-1/2}$, and phase responses $\phi_s$ (of *Equation 5* in Materials and Methods), are those of a low-pass (LP) filter controlled by a single parameter $\tau$. As $\tau$ is increased, the cut-off frequency of the LP filter becomes smaller and the phase shift increases in the lower frequencies (*Appendix 1—figure 1A*).

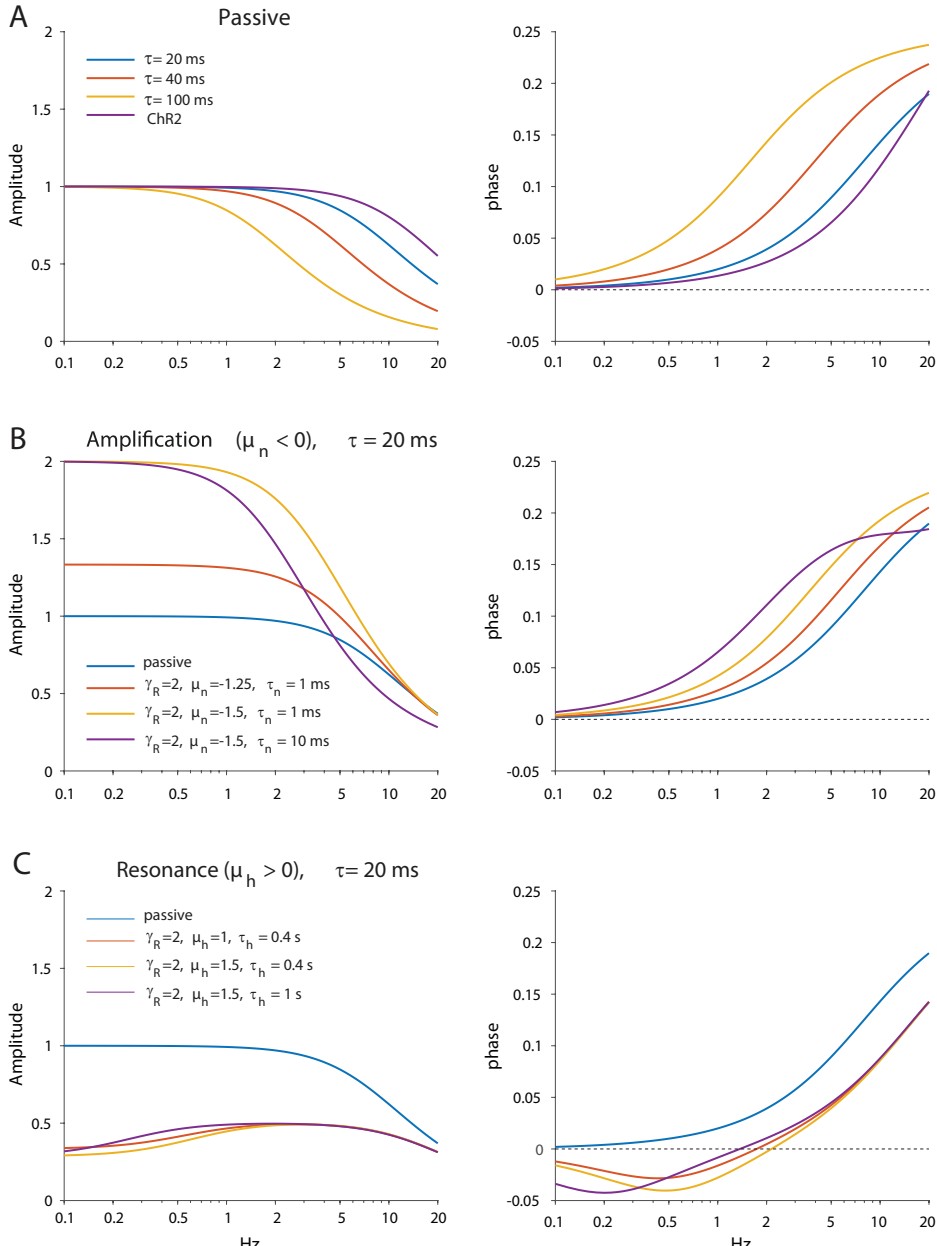

**Appendix 1—figure 1.** Dependence of amplitude and phase responses on parameters of the quasi-linear model. (**A**) Passive dendrites are controlled by a single parameter $\tau$. Amplitude and phase responses for typical values of the empirical alpha function used to model the ChR2 response ($\tau_r$=2 ms, $\tau_d$=11.5 ms) are is shown in purple for comparison. (**B**) Adding amplification increases the low frequency amplitude and phase response. (**C**) Adding resonance reduces the low frequency amplitude response and introduces negative phases in the low frequency phase response.

Next we introduce the amplifying current. We immediately see the boosting of the lower frequencies in the amplitude response as $\mu_n$ is made more negative. Here too, increasing $\tau_n$ shifts the cut-off to lower frequencies and increases the phase shift in the lower frequencies (**Appendix 1—figure 1B**). Note that the amplifying current can only increase the phase shifts.

When introducing a restorative current a resonance is created. While the resonance can be observed in the amplitude response as the formation of a non-zero maximal amplitude, *it is much more robustly* observed in the negative lobe that forms in the phase response at low frequencies (**Appendix 1—figure 1C**). Increasing $\mu_h$ deepens the negative lobe and shifts the zero-crossing (and the resonance peak) to a higher frequency. In contrast, increasing $\tau_h$ shifts the zero crossing

(and resonance) to a lower frequency. Note that for self-consistency when adding a nonlinear conductance, $\gamma_R$ must also be increased above 1 (**Goldberg et al., 2007**; **Remme and Rinzel, 2011**).

## Modeling dendritic optogenetic perturbation

When modeling the optogenetic activation of the dendrite we form a cascade of two filters: that of the ChR2 (**Equation 6b** in Materials and Methods, **Appendix 1—figure 1A**, "purple") and that of the dendrite. The parameters of ChR2 kinetics are based on our measurement of the rise ($\tau_r$) and decay ($\tau_d$) time constants (of the alpha function) that we fit to the data (**Figure 3—figure supplement 1**). However, the fact that the estimate of $\tau_d$ is systematically larger for the full-field illumination relative for the proximal illumination must result from the cable properties of the dendritic arbor (**Tiroshi and Goldberg, 2019**). This means that the underlying decay kinetics must be shorter than the decay time recorded under both proximal and full-field illumination. Therefore, for fitting the dendritic model, we chose $\tau_d = 10$ ms and $\tau_r = 0.2$ ms as representative values, both of which are in agreement with the literature on ChR2 kinetics in expression systems (**Nagel et al., 2003**).

An additional concern about the ChR2 kinetics is that they may have a longer time scale effect on our estimates of the amplitude and phase responses. To alleviate this concern we compared the phase estimates attained when sweeping the frequencies from low-to-high to sweeping them from high to low, and found that they do not differ (**Appendix 1—figure 2**).

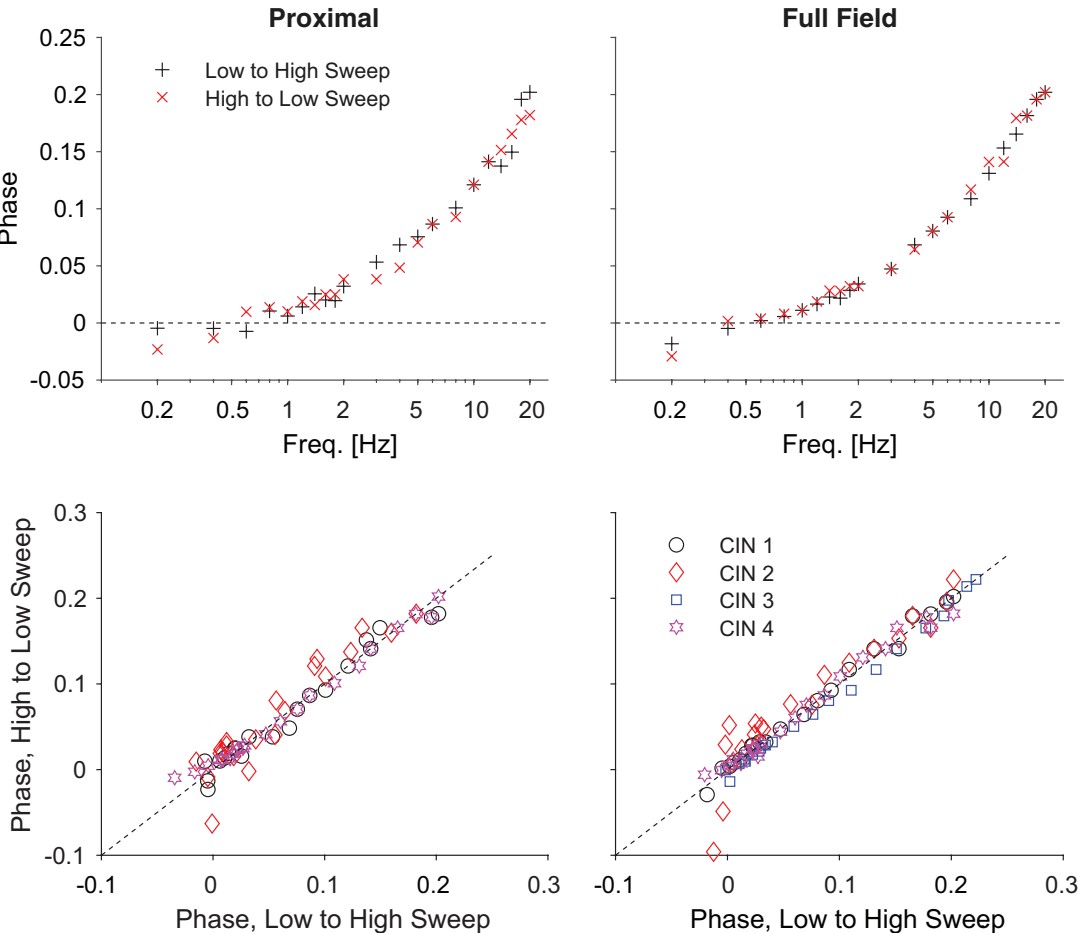

**Appendix 1—figure 2.** Reversing the optogenetic frequency sweep does not affect the phase estimates. Top: estimates of the phases as a function of frequency for one neuron for increasing (black) and decreasing (red) frequencies for proximal and full-field illumination. Bottom: Scatter plot of phases recorded for decreasing *vs.* increasing frequencies (for 4 cells) shows that the values cluster around the diagonal. Holding potential: −70 mV.

The dendritic amplitude response is given by $\left(1 - e^{-pr}\right)\left(p^2 + q^2\right)^{-1/2}$ (**Tiroshi and Goldberg, 2019**) where $r$ is the length of the dendrite being illuminated (in units of dendritic space constants, **Appendix 1—figure 3A**). Illumination of a longer extent of the dendrite results in a larger phase shift

in the higher frequencies of the phase response (*Appendix 1—figure 3B*), as seen in the present study (*Figure 3C and D*), and in our previous study of dendrites of GABAergic neurons in the substantia nigra pars reticulata (*Tiroshi and Goldberg, 2019*). Importantly, for a given positive value of $\mu_h$, when more of the dendrite is illuminated the negative lobe becomes larger (*Appendix 1—figure 3B*), in contrast to what we found in the case of the CINs (*Figure 3D*). This mathematical fact, strengthens our conclusion that in order to recapitulate the empirical observation of a smaller negative lobe (*Figure 3D*), the density of the restorative current (e.g., the HCN current) must decrease when more of the dendritic membrane is being illuminated.

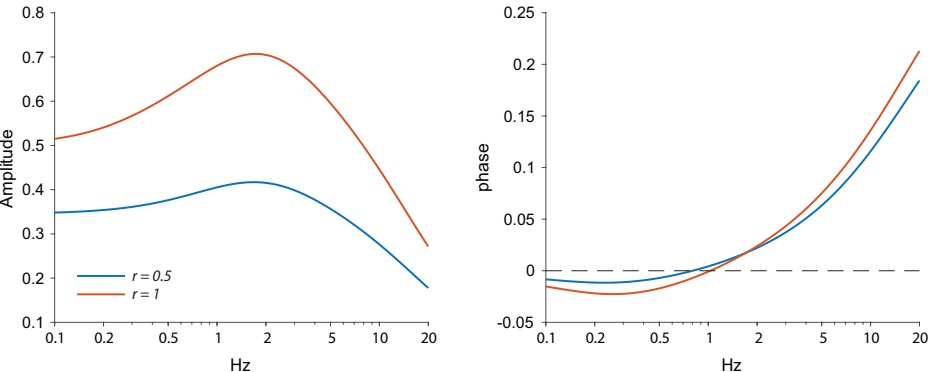

**Appendix 1—figure 3.** Amplitude and phase response arising from the ChR2 kinetics and a quasi-linear dendrite. A homogeneous distribution of quasi-linear properties results in a stronger resonance when a larger portion of the dendrite ($r=1$ vs. $r=0.5$) is illuminated, which can be seen as a sharper peak and a negative phase region with more negative phases. Other parameters: $\gamma_R=2$, $\tau=20$ ms, $\mu_n = -1.5$, $\mu_h=2$, $\tau_n=1$ ms, $\tau_h=0.8$ s, $\tau_r=0.2$ ms, $\tau_d=10$ ms.

Our model thus contains 6 free parameters ($\tau_r$ and $\tau_d$ were not treated as free parameters, as explained above) for the somatic perturbations (*Figures 1 and 2*) and one additional parameter $r$ for the optogenetic dendritic perturbations (*Figure 3*) (see *Table 1* in main text). Of course, the whole purpose of the model is to provide a more generalized method to characterize the impact of nonlinearities on dendritic integration, thereby reducing the number of free parameter to only this handful. Despite the significant dimensional reduction achieved by neglecting the fine details of the biophysics and kinetics of each of the channels that give rise to the nonlinearity (as well as the detailed dendritic morphology), we nonetheless attain several free parameters, which could support multiple viable fits to the same data, and might lead to overfitting of the data. To address this issue, we approached the model fitting as follows.

First, because (a) the phase response can be fit more robustly than the amplitude response (the phase response does not depend strongly on the intensity of the stimulus); (b) the negative phase region is a more robust measures of resonances than the amplitude (e.g, compare the left and right panels in *Figure 1C*); and (c) the error bars on the phase are much tighter – we fit our parameters based solely on the phase responses (In *Figures 1C, 2B and D*, *Figure 3—figure supplement 2*). Consequently, we fit a single parameter for the amplitude of the impedance curves using the parameters of the fit attained from the phase shifts.

Second, we only included parameters when they were necessary. Based on the empirical data demonstrating that (TTX-sensitive) NaP currents produce no negative lobes in the phase response, but nevertheless boost responses at –55 mV, we fit those data (e.g., *Figure 2B*) with a single amplifying nonlinearity. Only for the HCN currents that empirically gave rise to the negative lobe (e.g., *Figures 2D and 3D*) did we use the full model (i.e., 6 and 7 free parameters, respectively).

Third, we conducted least-square curve-fitting (Matlab, Mathworks) by restricting the values of the parameters so that the reflect physiologically reasonable values (*Appendix 1—table 1*), particularly for the membrane, NaP and HCN time constants (*Goldberg et al., 2007*).

The two following qualitative findings gave us confidence in our approach, and in our conclusions regarding the dendritic distribution of nonlinearities. First, the $r$ extracted from the full field configuration was always larger than the $r$ extracted for the proximal illumination, indicating that the measurements were sensitive to whether only a proximal region of the CIN was illuminated or its

entire dendritic field. Second, the $\mu$'s extracted for the full field illumination were smaller in absolute magnitude than those extracted for the proximal illumination, strengthening the conclusion that the channels that give rise to them are expressed primarily proximally.

In summary, while the quasi-linear model gives rise to multiple parameters, it is still a significantly dimensionally-reduced representation of dendritic morphology and filtering. Careful use of parameter fitting can be used to extract the qualitative properties of the dendrites, including regarding the localization of nonlinearities.

**Appendix 1—table 1.** Parameter ranges and initial guesses for the quasi-linear dendritic model fits.

| Parameter | $\mu_n$ | $\mu_h$ | $\tau_n$ (ms) | $\tau_h$ (s) | $\gamma_R$ | $\tau$ (ms) | $r$ |
|---|---|---|---|---|---|---|---|
| min | −5 | 0 | 0.1 | 0.01 | 1 | 10 | 0 |
| max | 0 | 5 | 100 | 10 | 10 | 100 | 2 |
| initial guess | −1 | 1 | 20 | 1 | 2 | 50 | 1 |

