## [Editor Report]

This manuscript addresses the cellular and dendritic physiology of cholinergic interneurons in the striatum. The authors use a creative integration of electrophysiology and optical methods to investigate this distinctive cell type, which is critically important at the intersection of motivated behavior and disease. They uncover a mechanism through which two separate active conductances – the hyperpolarization-activated h-current (HCN) and the persistent sodium current (NaP) – act in concert to selectively boost synaptic input from the thalamus onto proximal dendrites of cholinergic interneurons.

---

## [Decision Letter]

**Decision letter after peer review:**

Thank you for submitting your article "Non-uniform distribution of dendritic nonlinearities differentially engages thalamostriatal and corticostriatal inputs onto cholinergic interneurons" for consideration by *eLife*. Your article has been reviewed by 2 peer reviewers, and the evaluation has been overseen by a Reviewing Editor and John Huguenard as the Senior Editor. The following individual involved in review of your submission has agreed to reveal their identity: Scott Owen (Reviewer #1).

Essential revisions:

1, The use of ChR2 to induce the ZAP protocol. This is a central part of the story. The inactivation properties of the opsin should be discussed and it should even be incorporated in the model and tested in a few control experiments.

2. There is a disconnect between the first 4 figures and the last two figures. Regarding the primate data, the link to the slice work should be strengthened.

3. New physiology or modeling should be done to explain why cortical inputs to distal dendrites are not boosted as they pass through the proximal dendrite (Figure 5).

4. Both reviewers made detailed suggestions on additional edits and discussion points. Please take a close look at the reviewers' comments below, and incorporate these suggestions in the revision.

*Reviewer #1 (Recommendations for the authors):*

The authors should be congratulated on an important and well executed study. This scope of this manuscript seems to be a good fit for the readership at *eLife*, provided the concerns raised in the public section are adequately addressed.

To address the first major concern raised in public comments, regarding why boosting of proximal dendrites does not affect input from distal dendrites, additional text should be added to the Results and Discussion sections. Specifically, the authors should address whether this is a valid concern or a possible misconception, and how this can be resolved. This is an exceptionally important point, because without an adequate explanation, it is hard to understand how Figures 5 and 6 belong in the same manuscript as Figures 1-4.

To address the second major concern, regarding potential over-fitting of the model, the manuscript would benefit from additional tables and analysis. This description should include tables describing all fixed and free parameters that are fed into the model in Equations 1-4, and how fixed parameters were determined. In addition, a figure illustrating graphically how these parameters contribute to the plot fits would be invaluable. Which parameters contribute to saturation of phase drift at high and low frequencies? What sets the slope of phase drift? How many parameters had to be changed to allow a single model to fit both the simple, monotonically rising curve shape in Figure 3, and the very complex, multi-phasic curve in Figure 2B. In particular, how do these parameters interact with one another and how do those interactions affect the confidence with which results can be interpreted? i.e. how do we know that there are not other very different solutions to this model that provide equivalently good fits but point to very different physiological interpretations? A handful of well-chosen plots, following a format of the curves in Figures 1-3, but demonstrating which features of these curves are altered by specific model parameters, could be far more informative than any extensive discussion in text.

To address the third major concern, additional control experiments are likely required. The degree of inactivation of ChR2 is likely a function of light power and available channel population in the membrane, and therefore has to be measured empirically under specific experimental conditions. In the public comments, two experiments are suggested (directly measuring ChR2 inactivation with equivalent light power, and running the ramp backwards). Although it seems feasible to do both experiments in the same preparation, either one of these would likely be sufficient if both measurements cannot be made.

*Reviewer #2 (Recommendations for the authors):*

I have just a few comments for improving the paper, listed below in no particular order.

1. The zap protocol presented in the first figures has been used previously for CINs (Beatty 2015) and gave somewhat different resonance peaks. It would be interesting to discuss potential reasons for these differences.

2. Using the ZAP protocol by current/voltage somatic injection is likely to result in very different behavior than optogenetic entrainment due to the biophysical properties of ChR2 itself. ChR2 has its own activation and inactivation time constants which will affect the currents recorded during the ZAP protocol. Indeed, the phase-shift curves in Figure 3 are very different from those in figures 1 and 2.

3. The analysis of the backpropagation of action potentials into the CIN dendrites relies on the calcium responses presented in Figure 4D, however there is no direct measurement of the invasion of the AP and the measurement could also reflect the density of certain calcium channels and not only that of NaP ones. While performing dendritic electrophysiological recordings is labor intensive and may indeed be outside the scope of this study, this issue should be discussed. Also, the variability in the dendritic calcium responses presented in 4D is huge, and it is difficult to assess the statistical validity of the observed decay in response amplitude. Were there no responses beyond ~120 microns?

4. The synaptic input from optogenetic activation of cortical and thalamic input generated synaptic responses in the presence of AMPA, NMDA, GABAA, and GABAB blockers (according to the Methods section). What was the nature of these responses if they were not AMPA/NMDA mediated? If a different set of blockers was used it should be mentioned. Were the responses purely monosynaptic? This should be tested using TTX/4AP combination, as in Petreanu et al., (2009). If no blockers were used in these particular experiments, could there be polysynaptic interactions in addition to the monosynaptic responses?

5. While the thalamic inputs were activated by viral injections in the PfN of Vglut2-Cre mice, the cortical activation relies on the ChR2 reporter in Thy1 mice. Is ChR2 expressed only in cortical cells? Which other inputs may be activated? Is there no thalamic labeling in Thy1-ChR2 mice?

6. Could ranolazine have presynaptic effects on the optogenetic stimulation of axons? This could be checked by assessing changes in synaptic release properties by PPR or train optogenetic protocols.

7. The entrainment of TANs and MSNs to slow wave oscillations is interesting and the difference from SPNs is striking. There is a discrepancy with previous papers showing a very strong modulation of MSNs during cortical slow-wave oscillations (Stern et al., (1997, 1998), Reig and Silberberg (2014)). Intracellular (and whole-cell) recordings of CINs during slow-wave oscillations also showed various degrees of entrainment of the membrane potential to the cortical oscillation but to a lesser degree than MSNs (Schulz et al., (2011), Reig and Silberberg (2014)). This does not necessarily mean that the input is cortical since thalamus also displays slow-wave oscillations under the same conditions. Moreover, natural slow wave sleep is different from slow wave anesthesia-induced sleep. These differences would be interesting to discuss.

8. Is the entrainment of TANs to slow-wave oscillations stronger than to other "sleep frequencies"? How locked are they to the higher frequencies in other sleep stages?

[Editors’ note: further revisions were suggested prior to acceptance, as described below.]

Thank you for resubmitting your work entitled "Non-uniform distribution of dendritic nonlinearities differentially engages thalamostriatal and corticostriatal inputs onto cholinergic interneurons" for further consideration by *eLife*. Your revised article has been evaluated by John Huguenard (Senior Editor) and a Reviewing Editor.

The manuscript has been improved but there are some remaining issues that need to be addressed, as outlined below:

Further experiment or simulation is needed to address the discrepancy between the modeling in Appendix 2 and the physiology results in Figure 5.

*Reviewer #1 (Recommendations for the authors):*

The authors have done an admirable job of responding to most of the major comments through a combination of new experiments, modeling, and conceptual insight. Their efforts are especially thorough with respect to questions regarding how the kinetics of ChR2 may influence the interpretation of results from the Chirp stimulus protocol. The modified manuscript is substantially clarified and improved.

However, the new manuscript still falls short of convincingly addressing one of the primary conceptual questions raised in the previous review: Why are cortical inputs onto distal dendrites not boosted as they pass through a proximal dendrite? The authors offer an explanation (that the NaP current is voltage-dependent and nonlinear), which appears well suited to explain the new modeling results in Appendix 2. However, there appears to be a fundamental discrepancy between the modeling in Appendix 2 and the physiology results in Figure 5. In the modeling data (Appendix 2), the cortical EPSP is far smaller at the soma than the thalamic EPSP (presumably due to attenuation/leak over the length of the dendrite?). This reduced amplitude of the cortical EPSP as it passes through the proximal dendrite seems well suited to account for the lack of boosting observed in the model. However, in the physiology data (Figure 5), the cortically evoked EPSP is equivalent size or larger at the soma than the thalamically evoked EPSP. Wouldn't this mean that, in this experiment, the cortical EPSP is at least as large as the thalamic EPSP when it passes through the proximal dendrite?

If cortical and thalamic EPSPs are each driving equivalent depolarization of the proximal dendrite in Figure 5, how is it that the cortically evoked EPSP does not experience the same non-linear boosting as the thalamic EPSP?

In order to adequately explain the physiology data in Figure 5, the modeling in Appendix 2 should compare evoked inputs from different sources that elicit equivalent EPSP amplitude at the soma. If this model cannot explain differential boosting of inputs that elicit equivalent EPSP amplitudes at the soma, some alternate interpretation should be provided for the surprising physiological result in Figure 5.

*Reviewer #2 (Recommendations for the authors):*

My comments have been addressed and I congratulate the authors for a very interesting paper.

Just for curiosity, what is the advantage in using discrete frequency transitions rather than a continuous frequency increase in the ZAP protocol?

---

## [Author Response]

Essential revisions:1, The use of ChR2 to induce the ZAP protocol. This is a central part of the story. The inactivation properties of the opsin should be discussed and it should even be incorporated in the model and tested in a few control experiments.

Both reviewers (and editors) were absolutely right about this point! We have now performed several experiments that demonstrate that the phase response at large frequencies (Figure 3) is dominated by the contribution of the filtering properties of channelrhodopsin-2 (ChR2). To demonstrate this, we recorded the somatic currents generated by proximal or full-field 470 nm illumination (Figure 3—figure supplement 1A) and found that the empirical rise and decay time constants of the α-function-shaped response (Equation 1 in revised manuscript), when plugged into the Fourier transform of this function (Equation 6 in the revised manuscript) closely fit the empirical phase responses at high frequencies (Figure 3—figure supplement 1C). In contrast, the ChR2 cannot account for the observed negative phases (e.g., Figure 3D).

In light of this new observation, we realized we must include a model for the amplitude and phase response of the ChR2 kinetics in our fits of the quasi-linear approximation. Because we now have a cascade of two filters (first ChR2 kinetics and then dendritic morphology and nonlinearity), the ChR2 phase is added to the quasi-linear phase response. Of the two parameters of the ChR2 kinetics – the rise and decay times – the latter is the one that dominates the contribution to the phase. The finding that the empirical decay time for the full illumination (~14 ms) is systematically larger (Figure 3—figure supplement 1B) than for the proximal illumination (~11.5 ms), can only be attributed to the difference in the electrotonic range being illuminated (as we showed in our previous publication, Tiroshi et al., *PLoS Comp Biol* 2019). This means that the underlying decay kinetics of the ChR2 must be shorter than both these estimates. So in our fits, we chose an underlying decay time constant of 10 ms, which is consistent with the literature.

In addition to reconciling the huge difference between the phase delays elicited optogenetically and those elicited with the somatic voltage perturbations (Figures 1-2) – which was an issue raised by Dr. Owen – adding ChR2 kinetics also helped some other results fall into place. The order-of-magnitude discrepancy that we had estimated in the initial submission between the µ parameters that characterize the somatic experiment (Figures 1-2) *vs.* those that characterize the dendritic experiments (Figure 3) vanished. Now the µ parameters are on the same order as those extracted from the somatic experiments. We also found that the estimate of the electrotonic range being illuminated at –55 mV (Figure 3C) and –70 mV is now more consistent than we had previously found.

With respect to slower ChR2 kinetics that could come into play (e.g., inactivation), we followed Dr. Owen’s recommendation and compared the estimated phase response to an increase sweep of frequency and a decreasing one, but found no difference (Appendix 1–figure 2).

The most important outcome of adding ChR2 dynamics into our model fit, came from the fact that the ChR2 model fit the data at –55 mV “too well” which, at first sight, seemed to preclude boosting by persistent sodium current. This forced us to conduct another experiment where we directly tested the effect of blocking NaP with tetrodotoxin (TTX) on boosting at –55 mV. To our delight, the result of this experiment provided evidence – that is independent of the model fit – for proximal-only boosting. We found that TTX had an opposite effect on the amplitude response when illumination was proximal vs. when it was full-field. Whereas TTX increased the amplitude response in the case of full-field illumination (presumably by reducing the dendritic leak, because NaP is a leak current in the subthreshold range), TTX reduced the amplitude response when illumination was proximal, which is indicative of boosting, just as we found with somatic voltage perturbations. This finding significantly strengthens our manuscript, because it provides new model-independent pharmacological evidence of proximal-only boosting, which we did not provide in the original submission that relied strictly on parameter fitting to reach this conclusion.

2. There is a disconnect between the first 4 figures and the last two figures. Regarding the primate data, the link to the slice work should be strengthened.

As pointed at the end of the response to the previous comment, we now provide new model-independent evidence of proximal-only boosting. This of course strengthens our indirect evidence from calcium imaging of back-propagating action potentials (bAPs, Figure 4) regarding the proximal localization of the voltage-activated (NaV) sodium channels underlying the NaP current. It also sets the stage for the experiment in Figure 5, where we show that only proximal thalamic EPSPs are ranolazine-sensitive.

In addition, Reviewer #2 requested several controls and additional data to Figure 5, which we have now done: we demonstrate that the PSPs recorded in response to optogentic activation of cortical fibers in the Thy1-ChR2 mouse and PfN-axons in the Vglut2-cre mouse are excitatory and monosynaptic, and provided a confocal image of the EYFP in the PfN, that shows that it is present in (nominally cortical) fibers but not in cell bodies, so that there is no contamination with thalamic activation when 470 nm light is used in the Thy1-ChR2 mice (as shown previously in Gradinaru et al., *Science* 2009). We also tested, as requested by Reviewer #2, whether ranolazine could have a differential presynaptic effect on the two excitatory inputs to CINs (for cortex in Thy1-ChR2 mice and PfN in the Vglut2-cre mice) and found that it had no impact on paired-pulse ratios.

We also followed a suggestion of Reviewer #2 regarding strengthening the link to the primate data. We have now included an additional panel in Figure 6, that shows that TANs (comprised mostly of CINs) do not show time locking to higher-frequency sleep spindles. This finding strengthens the conclusion that TAN/CINs exhibit a preference for lower-frequency inputs, which *could* be related to the resonant properties we studied. While there is a huge jump here, it is certainly worth noting these results in light of our finding, as it provides a potential in vivo correlate of our in vitro findings.

3. New physiology or modeling should be done to explain why cortical inputs to distal dendrites are not boosted as they pass through the proximal dendrite (Figure 5).

We have now added an Appendix (#2) to the manuscript to discuss this point. As pointed out in Appendix 2 and echoed in the main text (and Dr. Owen is absolutely correct here), if the dendrite were strictly linear (passive or quasi-linear) then scaling the input would precisely scale the output. However, because the NaP current is voltage-dependent and nonlinear, for large enough inputs there will be a considerably stronger boosting of proximal inputs in comparison to distal ones, when the nonlinearity is restricted to a proximal region. We demonstrate this result by simulating a dendrite with the NaP current expressed only up to some distance from the soma, and then the dendrite is passive beyond that point. This mechanism provides a feasible explanation for how proximal thalamic inputs can be boosted and hence ranolazine-sensitive, whereas distal cortical inputs are not boosted and hence ranolazine-insensitive (Figure 5).

4. Both reviewers made detailed suggestions on additional edits and discussion points. Please take a close look at the reviewers' comments below, and incorporate these suggestions in the revision.

We have addressed all the reviewers individual requests as described below.

Reviewer #1 (Recommendations for the authors):The authors should be congratulated on an important and well executed study. This scope of this manuscript seems to be a good fit for the readership at eLife, provided the concerns raised in the public section are adequately addressed.To address the first major concern raised in public comments, regarding why boosting of proximal dendrites does not affect input from distal dendrites, additional text should be added to the Results and Discussion sections. Specifically, the authors should address whether this is a valid concern or a possible misconception, and how this can be resolved. This is an exceptionally important point, because without an adequate explanation, it is hard to understand how Figures 5 and 6 belong in the same manuscript as Figures 1-4.

Thank you. We have addressed these concerns above.

To address the second major concern, regarding potential over-fitting of the model, the manuscript would benefit from additional tables and analysis. This description should include tables describing all fixed and free parameters that are fed into the model in Equations 1-4, and how fixed parameters were determined. In addition, a figure illustrating graphically how these parameters contribute to the plot fits would be invaluable. Which parameters contribute to saturation of phase drift at high and low frequencies? What sets the slope of phase drift? How many parameters had to be changed to allow a single model to fit both the simple, monotonically rising curve shape in Figure 3, and the very complex, multi-phasic curve in Figure 2B. In particular, how do these parameters interact with one another and how do those interactions affect the confidence with which results can be interpreted? i.e. how do we know that there are not other very different solutions to this model that provide equivalently good fits but point to very different physiological interpretations? A handful of well-chosen plots, following a format of the curves in Figures 1-3, but demonstrating which features of these curves are altered by specific model parameters, could be far more informative than any extensive discussion in text.

We believe that with the additional table, with the dedicated Appendix 2, and the discussion of the problem of over-fitting we have addressed this concern, as explained above. Additionally, the new pharmacological demonstration of the proximal localization of the NaP currents, in a sense, steals the thunder from the model, which is now less important for the main thrust of the paper. But nevertheless, it is useful in estimating the localization of the HCN current, and provides a framework to understand the use of optogenetics in activating neurons post-synaptically in order to study their dendritic nonlinearities.

To address the third major concern, additional control experiments are likely required. The degree of inactivation of ChR2 is likely a function of light power and available channel population in the membrane, and therefore has to be measured empirically under specific experimental conditions. In the public comments, two experiments are suggested (directly measuring ChR2 inactivation with equivalent light power, and running the ramp backwards). Although it seems feasible to do both experiments in the same preparation, either one of these would likely be sufficient if both measurements cannot be made.

As explained above we have conducted these experiments and included the new results.

Reviewer #2 (Recommendations for the authors):I have just a few comments for improving the paper, listed below in no particular order.1. The zap protocol presented in the first figures has been used previously for CINs (Beatty 2015) and gave somewhat different resonance peaks. It would be interesting to discuss potential reasons for these differences.

We have now discussed this point, as follows:

“It also seems that the location of the peak amplitudes estimated by Beatty and collaborators (2015) is perhaps 1 Hz larger than the location of the peaks in our study. However, this difference could result from the fact that they used a chirp stimulation where the frequency increases continuously, whereas we used several periods of perfect sine waves with a discrete set of frequencies. Moreover, it can be shown mathematically that for the quasi-linear model the zero crossing in the phase response occurs at a slightly smaller frequency than the peak in the amplitude response, so this may also contribute to the impression that the resonance frequency observed in our experiments is slightly different from that observed in Beatty and collaborators (2015).”

2. Using the ZAP protocol by current/voltage somatic injection is likely to result in very different behavior than optogenetic entrainment due to the biophysical properties of ChR2 itself. ChR2 has its own activation and inactivation time constants which will affect the currents recorded during the ZAP protocol. Indeed, the phase-shift curves in Figure 3 are very different from those in figures 1 and 2.

Touché, to you as well. As discussed extensively above both in the “Essential Revisions” and in response to Dr. Owen’s same critique, this issue is now resolved with new experiments and new analysis that include a model for the phase delays generated by the ChR2 kinetics. This has resulted in major changes to the text and figures.

3. The analysis of the backpropagation of action potentials into the CIN dendrites relies on the calcium responses presented in Figure 4D, however there is no direct measurement of the invasion of the AP and the measurement could also reflect the density of certain calcium channels and not only that of NaP ones. While performing dendritic electrophysiological recordings is labor intensive and may indeed be outside the scope of this study, this issue should be discussed. Also, the variability in the dendritic calcium responses presented in 4D is huge, and it is difficult to assess the statistical validity of the observed decay in response amplitude. Were there no responses beyond ~120 microns?

We have now added a comment about the possibility that bAPs amplitude represent Cav channel densities, rather than Nav channel density, as follows:

“Because strictly speaking a change in Ca^2+^ transients could result from a change in the concentration of voltage-activated Ca^2+^ channels, future work can verify this by directly testing where locally puffed TTX attenuates the bAP (perhaps even with the use of voltage sensitive probes).”

I have conducted 2PLSM measurements of bAPs in 275-300 µm slices in my own lab and previously in the laboratory of Jim Surmeier. In both labs, I was rarely able to get measurements beyond 120 or at most 150 µm. The variability in the responses is indeed large, so in order to lend statistical validity to our conclusion that the calcium transient decay distally, we have complemented our analysis with an additional statistical test conducted on the same set of data points, where we compare amplitudes of calcium transient that are more proximal vs. more distal than 70 µm, as explained in the manuscript “Additionally, in 7 CINs (*N = 5 mice*) in which we had measurements of *∆F/F_o_* at both proximal and distal (>70 µm) locations, we found the median distal signal was significantly lower than the proximal one by 29% (P = 0.047, Wilcoxon rank sum test).” We hope this makes this highly variable data slightly more palatable.

Finally, as mentioned several times above, our new experiment in which TTX had an opposite effect on the amplitude response elicited with proximal *vs.* full-field illumination, provides incontrovertible data that Nav channels density/function is different proximally vs. distally. So in a sense, our previous conclusion from the bAP data is buttressed by this new experiment. Thus, the main thrust of the paper relies currently less on the bAP data, which are nonetheless supportive and valuable IMHO.

4. The synaptic input from optogenetic activation of cortical and thalamic input generated synaptic responses in the presence of AMPA, NMDA, GABAA, and GABAB blockers (according to the Methods section). What was the nature of these responses if they were not AMPA/NMDA mediated? If a different set of blockers was used it should be mentioned. Were the responses purely monosynaptic? This should be tested using TTX/4AP combination, as in Petreanu et al., (2009). If no blockers were used in these particular experiments, could there be polysynaptic interactions in addition to the monosynaptic responses?

Apologies. This was an oversight. We had AMPAR, NMDAR, GABAaR, GABAbR (and also mAChR and nAChR) blockers in all experiments except in the optogenetic activation of synaptic inputs in which the everything except the AMPAR blocker were used. We have now used the method from Petreanu 2009 (Figure 5—figure supplement 1A) to verify that both the PfN inputs and the cortical inputs we activated were monosynaptic (and excitatory given the cocktail of GABAR we used). All CINs from Thy1-ChR2 mice (*n = 8*, *N = 2* mice) met the Petreanu criterion, and all but one (6/7) from the Vglut2-cre mice (*N = 2* mice).

5. While the thalamic inputs were activated by viral injections in the PfN of Vglut2-Cre mice, the cortical activation relies on the ChR2 reporter in Thy1 mice. Is ChR2 expressed only in cortical cells? Which other inputs may be activated? Is there no thalamic labeling in Thy1-ChR2 mice?

A previous paper (Gradinaru et al., *Science* 2009) reported no expression of ChR2 in PfN neurons in the Thy1-ChR2 mice used. In accordance with that report, we present confocal images of the PfN showing that there is ChR2 expression in (nominally cortical) fibers but not in cell bodies (Figure 5—figure supplement 1B). While based on our own data we cannot strictly rule out another source of monosynaptic excitatory inputs other than cortical inputs in the Thy1-ChR2 mice, such as the monosynaptic glutamatergic input to CINs from the pedunculopontine nucleus (PPN, Assous et al., *J Neurosci* 2019), the Gradinaru et al., (2009) study reported that PPN neurons do not express ChR2. Importantly, we know the input in this mouse are not contaminated by PfN inputs, and that these mice have been widely used by various labs as mice whose striatal innervation arises mostly from excitatory cortical origins. Finally, ranolazine did not affect the glutamatergic EPSPs in this mouse, so, hypothetically, even if there is another source of glutamatergic input it behaves like cortical inputs vis-à-vis postsynaptic boosting by CINs.

6. Could ranolazine have presynaptic effects on the optogenetic stimulation of axons? This could be checked by assessing changes in synaptic release properties by PPR or train optogenetic protocols.

We measured PPRs in both mice (the values were consistent with our own previous publication, Aceves-Buendia et al., *EJN* 2019) and found that ranolazine had no effect on them (Figure 5—figure supplement 1C).

7. The entrainment of TANs and MSNs to slow wave oscillations is interesting and the difference from SPNs is striking. There is a discrepancy with previous papers showing a very strong modulation of MSNs during cortical slow-wave oscillations (Stern et al., (1997, 1998), Reig and Silberberg (2014)). Intracellular (and whole-cell) recordings of CINs during slow-wave oscillations also showed various degrees of entrainment of the membrane potential to the cortical oscillation but to a lesser degree than MSNs (Schulz et al., (2011), Reig and Silberberg (2014)). This does not necessarily mean that the input is cortical since thalamus also displays slow-wave oscillations under the same conditions. Moreover, natural slow wave sleep is different from slow wave anesthesia-induced sleep. These differences would be interesting to discuss.

We have now discussed this issue, as follows:

“Our findings in sleeping NHPs are at odds with various rodent studies that found that cortical slow-wave activity is weakly associated with the discharge of TANs in anesthetized rodents, and more strongly associated with SPN activity (Goldberg et al., 2003; Mahon, 2001; Reig and Silberberg, 2014; Schulz et al., 2011; Sharott et al., 2012; Stern et al., 1998). These differences may be attributable to species differences and/or differences between natural sleep and anesthesia, which may differ in the degree to which thalamic *vs.* cortical inputs engage striatal neurons. Moreover, the dendritic nonlinearities in SPNs (Plotkin et al., 2011) may be preferentially engaged during anesthesia, thereby causing SPNs to respond more strongly under anesthesia than during natural sleep.”

8. Is the entrainment of TANs to slow-wave oscillations stronger than to other "sleep frequencies"? How locked are they to the higher frequencies in other sleep stages?

We have now added another panel to Figure 6 showing that while TANs are time locked to slow-wave events they are not locked to sleep spindles. This result strengthens the hypothesis that the TANs’ preferential entrainment by slow-wave events (but not sleep spindles) could arise from the CIN’s membrane resonance in the same range.

[Editors’ note: further revisions were suggested prior to acceptance, as described below.]

The manuscript has been improved but there are some remaining issues that need to be addressed, as outlined below:Further experiment or simulation is needed to address the discrepancy between the modeling in Appendix 2 and the physiology results in Figure 5.Reviewer #1 (Recommendations for the authors):The authors have done an admirable job of responding to most of the major comments through a combination of new experiments, modeling, and conceptual insight. Their efforts are especially thorough with respect to questions regarding how the kinetics of ChR2 may influence the interpretation of results from the Chirp stimulus protocol. The modified manuscript is substantially clarified and improved.However, the new manuscript still falls short of convincingly addressing one of the primary conceptual questions raised in the previous review: Why are cortical inputs onto distal dendrites not boosted as they pass through a proximal dendrite? The authors offer an explanation (that the NaP current is voltage-dependent and nonlinear), which appears well suited to explain the new modeling results in Appendix 2. However, there appears to be a fundamental discrepancy between the modeling in Appendix 2 and the physiology results in Figure 5. In the modeling data (Appendix 2), the cortical EPSP is far smaller at the soma than the thalamic EPSP (presumably due to attenuation/leak over the length of the dendrite?). This reduced amplitude of the cortical EPSP as it passes through the proximal dendrite seems well suited to account for the lack of boosting observed in the model. However, in the physiology data (Figure 5), the cortically evoked EPSP is equivalent size or larger at the soma than the thalamically evoked EPSP. Wouldn't this mean that, in this experiment, the cortical EPSP is at least as large as the thalamic EPSP when it passes through the proximal dendrite?If cortical and thalamic EPSPs are each driving equivalent depolarization of the proximal dendrite in Figure 5, how is it that the cortically evoked EPSP does not experience the same non-linear boosting as the thalamic EPSP?In order to adequately explain the physiology data in Figure 5, the modeling in Appendix 2 should compare evoked inputs from different sources that elicit equivalent EPSP amplitude at the soma. If this model cannot explain differential boosting of inputs that elicit equivalent EPSP amplitudes at the soma, some alternate interpretation should be provided for the surprising physiological result in Figure 5.

Thank you for the overall positive assessment of our work. While I am convinced that the mechanism I proposed in the text and the previous Appendix 2 provides some insight into how proximal nonlinearity can preferentially boost proximal inputs, I concede that Dr. Owen has raised issues that would nevertheless force me to revert to hand-waving arguments, anyways. Therefore, I decided to keep it simple (and also shorten the already lengthy manuscript) and circumvent the need for the modeling in Appendix 2 altogether (so I’ve removed Appendix 2). Instead, I now use the comparison of the current findings (Figures 2-4) to findings we published previously about Q175 mice (Tanimura et al., 2016), to motivate the experiments presented in Figure 5. We now show that there is a consistent set of findings relating how far NaP extends into CIN dendrites and the ability to boost inputs in an NaP-dependent fashion: in control animals it seems that NaP expression terminates more proximally, and so only proximal PfN inputs are boosted by the amplification it generates proximally; whereas in Q175 mice, NaP extends the region of amplification farther out so that distal cortical inputs can be boosted by it.

Reviewer #2 (Recommendations for the authors):My comments have been addressed and I congratulate the authors for a very interesting paper.Just for curiosity, what is the advantage in using discrete frequency transitions rather than a continuous frequency increase in the ZAP protocol?

Thank you. We feel that with a continuous zap where frequency is constantly changing it’s really hard to determine and absolute phase relationship between input and output. We therefore prefer our method (which we also published previously, Tiroshi and Goldberg 2019) which presents 1 or more cycles of perfect sinusoids (at discrete frequencies) and then use the cross-correlation between the input and the output to determine the phase at each of the discrete frequencies.